# Longitudinal single-cell analysis of a myeloma mouse model identifies subclonal molecular programs associated with progression

Danielle C. Croucher [1,2], Laura M. Richards [1,2,6], Serges P. Tsofack[1,6], Daniel Waller [3], Zhihua Li[1], Ellen Nong Wei[1], Xian Fang Huang[3], Marta Chesi [4], P. Leif Bergsagel [4], Michael Sebag[3], Trevor J. Pugh [1,2,5,7✉] & Suzanne Trudel [1,2,7✉]

Molecular programs that underlie precursor progression in multiple myeloma are incompletely understood. Here, we report a disease spectrum-spanning, single-cell analysis of the Vκ*MYC myeloma mouse model. Using samples obtained from mice with serologically undetectable disease, we identify malignant cells as early as 30 weeks of age and show that these tumours contain subclonal copy number variations that persist throughout progression. We detect intratumoural heterogeneity driven by transcriptional variability during active disease and show that subclonal expression programs are enriched at different times throughout early disease. We then show how one subclonal program related to GCN2 stress response is progressively activated during progression in myeloma patients. Finally, we use chemical and genetic perturbation of GCN2 in vitro to support this pathway as a therapeutic target in myeloma. These findings therefore present a model of precursor progression in Vκ*MYC mice, nominate an adaptive mechanism important for myeloma survival, and highlight the need for single-cell analyses to understand the biological underpinnings of disease progression.

[1] Princess Margaret Cancer Centre, University Health Network, Toronto, ON, Canada. [2] Department of Medical Biophysics, University of Toronto, Toronto, ON, Canada. [3] Department of Medicine, McGill University, Montréal, QC, Canada. [4] Division of Hematology/Oncology, Mayo Clinic, Scottsdale, AZ, USA. [5] Ontario Institute for Cancer Research, Toronto, ON, Canada. [6] These authors contributed equally: Laura M. Richards, Serges P. Tsofack. [7] These authors jointly supervised this work: Trevor J. Pugh, Suzanne Trudel. ✉email: trevor.pugh@utoronto.ca; Suzanne.Trudel@uhn.ca

Multiple myeloma (MM) is a haematological malignancy characterized by the clonal expansion of immunoglobulin-secreting plasma cells in the bone marrow (BM)[1]. Despite improvements in treatment over the last two decades, MM remains incurable, with a median 5-year survival of 44%[2]. All patients with active MM progress through a preceding spectrum of precursor asymptomatic disease stages known as monoclonal gammopathy of undetermined significance (MGUS) and smouldering MM (SMM)[3,4]. Although the standard of care for MGUS and SMM patients is "watchful waiting", recent findings from randomized clinical trials have shown that early treatment can reduce the risk of progression to MM in a subset of precursor patients[5]. However, the molecular determinants of what constitutes high-risk precursor disease remain unclear. Current stratification approaches are based on clinical parameters that define tumour burden and not biological differences that may underlie progression. Thus, a better understanding of myeloma biology in the context of progression is needed to improve surveillance, clinical management, and treatment.

Previous attempts to unravel the molecular underpinnings of disease progression in myeloma have relied on the use of bulk genomic sequencing[6–9] and transcriptional analyses[10,11]. These studies have shown that the genomic landscape of precursor myeloma is similar to active disease but reveals a handful of aberrations that are more commonly identified in precursor patients that progress (i.e., high-risk precursor disease) including dysregulation of MYC, MAPK, and DNA repair pathways[7]. However, these studies lack the resolution to identify distinct malignant cell states, do not reveal the impact of genetic events such as copy number variations (CNVs) on gene expression, and are unable to characterize features of intratumoural heterogeneity. Instead, the recent application of single-cell sequencing technologies to study the myeloma disease spectrum has revealed significant molecular complexity and variability in the tumour ecosystem at all stages of disease[12,13]. For example, Ledergor et al. used single-cell RNA sequencing (scRNA-seq) to define transcriptional heterogeneity in the malignant cell compartment within and between patients with MGUS, SMM, and MM[12]. However, the authors did not relate these patterns back to disease progression, nor did their results validate therapeutic vulnerabilities. The challenge of defining drivers of progression in these studies may result from inter-patient heterogeneity, extensive genomic complexity, and tumour subclonality that characterizes myeloma[14–16]. Therefore, it is possible that the application of scRNA-seq to a genetically uniform cohort of subjects may provide the clarity needed to better understand the molecular mechanisms associated with progression.

The Vκ*MYC mouse model of MM allows for a controlled comparison across disease stages as malignant transformation is uniformly driven by constitutive activation of the MYC oncogene[17]. Vκ*MYC mice inevitably develop a progressive accumulation of clonal plasma cells in the BM that faithfully recapitulate the human disease. The genetic background of this model is also highly relevant as MYC rearrangements are found in half of human MM tumours[18–21], including SMM[19], and because MYC dysregulation is implicated in high-risk SMM progression[19,22]. Indeed, the natural history of Vκ*MYC disease progression is said to likely follow the early stages of a higher-risk precursor disease to a clinically-defined overt MM, including a transitory smouldering-like period[23]. Moreover, the disease can be monitored in mice indirectly by measuring monoclonal immunoglobulins (M-protein) in the blood. Thus, the Vκ*MYC model provides a unique opportunity to assess the molecular profiles of malignant cells from the earliest stages of myeloma, prior to serological detection of disease, which is typically not feasible in human studies.

Here, we use this genetically uniform tumour model to characterize the molecular features of malignant cells during myeloma disease progression over time. Using a single-cell approach, we explore sources of tumour heterogeneity longitudinally in the malignant cell compartment of Vκ*MYC tumours and characterize the relationship between subclonal genomic events and subclonal transcriptional programs. These analyses reveal that the malignant cell compartment of mice with early disease consists of multiple CNV-driven subpopulations and is enriched for transcriptional programs identified as subclonal in mice with active-MM. Moreover, pathway-level analyses revealed that one of these subclonal pathways relates to an adaptive program in malignant cells involving GCN2 and the integrated stress response pathway. Finally, we show how activation of this pathway is associated with disease progression in human myeloma patients and validate GCN2 experimentally as a promising therapeutic target in a subset of myeloma tumours. Thus, our study demonstrates how subclonal molecular programs can inform targeted therapeutic strategies in MM.

## Results

**A disease spectrum-spanning cohort of Vκ*MYC tumours.** To model the evolutionary stages of human MM disease progression, we established a cohort of Vκ*MYC mice on the C57BL/6/KaLwRij background strain (Fig. 1a and Supplementary Data 1), which has been shown to have an increased propensity for developing spontaneous monoclonal gammopathies[24,25]. The cohort included 5 Vκ*MYC mice without detectable disease (early-MM, EMM1-5, 27–33 weeks), 3 Vκ*MYC mice with intermediate disease (int-MM, IMM1-3, 49 weeks), and 7 Vκ*MYC mice with active MM (active-MM, AMM1-7, 61–74 weeks). We also included 3 C57BL/6/KaLwRij mice that served as age-matched controls (Control, Cont1-3, 55–72 weeks). Disease stage groups were defined based on serum M-protein levels (Fig. 1b) and age (Supplementary Data 1). While the distinction between early and intermediate disease based on these criteria was subtle, mice with active disease demonstrated a significantly higher serum M-protein compared to all other mice (Fig. 1b), supporting an exponential pattern of progression. To characterize malignant cell states throughout disease progression, we used droplet-based scRNA-seq and profiled unselected, red blood cell-depleted cells derived from femoral BM of mice (Fig. 1a). After removing low-quality cells (see "Methods" for details, Supplementary Fig. 1a), we considered expression profiles for 104,880 cells across 18 samples, with a median of 5,258 cells sequenced per sample (range: 973–10,245 cells). This included 13,296 cells from early-MM mice, 27,510 cells from int-MM mice, 44,463 cells from active-MM mice, and 19,611 cells from control mice. This scRNA-seq data set therefore provides a valuable resource for the MM community and for those who employ the Vκ*MYC mouse model.

To distinguish cell types within the BM, we used a combination of dimensionality reduction and unsupervised clustering to obtain an integrated map of 52 transcriptional clusters (Supplementary Fig. 1b). Cell lineages were then assigned to each cluster using SingleR[26], which annotates scRNA-seq data using a reference set with known labels. For our data set of BM cells, we used the ImmGen[27] reference comprised of bulk gene expression profiles from highly purified immune and hematopoietic cell populations (Supplementary Fig. 1c, d). This revealed 17,504 cells from the B cell lineage (Fig. 1c), which we further refined by scoring expression profiles using a BM plasma cell gene set generated by the Human Cell Atlas[28] (Fig. 1d). This, together with the expression of canonical marker genes Cd19 and Sdc1 (Fig. 1e), enabled discrimination of 10,344 B cells and 7,160 plasma cells in the BM of this cohort (Supplementary Data 1).

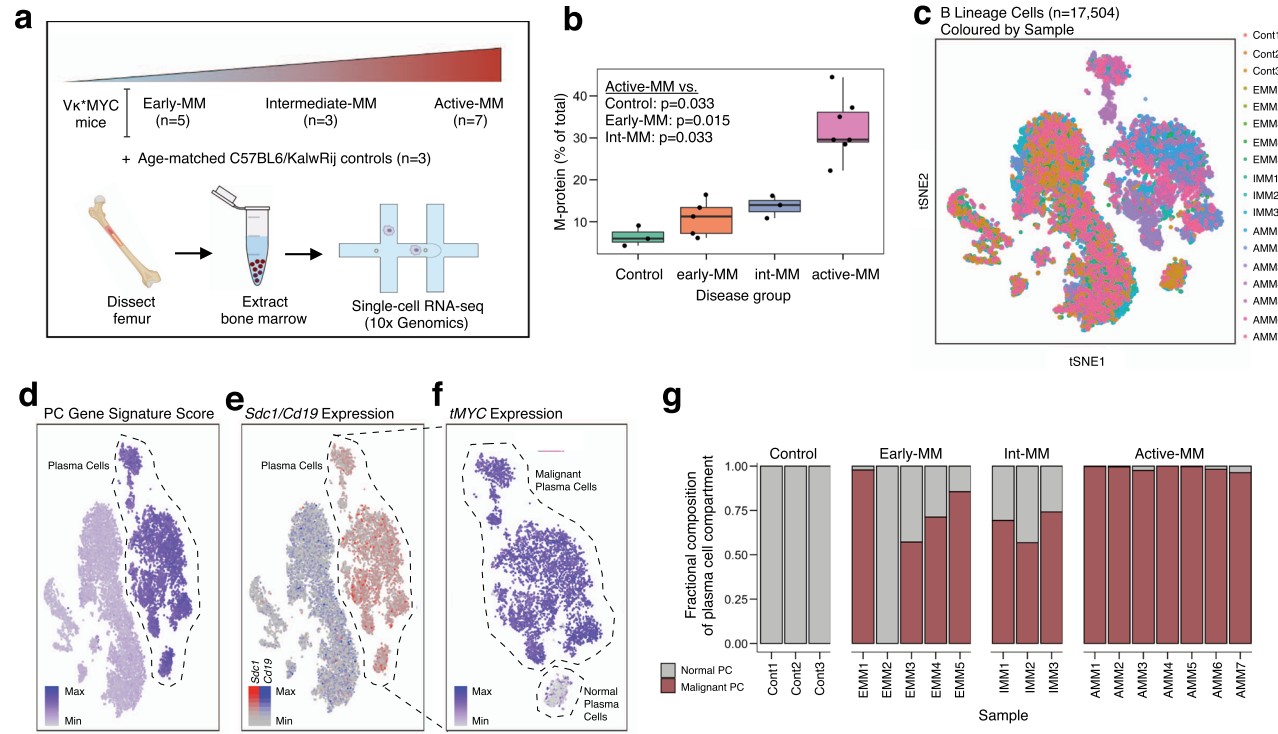

**Fig. 1 A single-cell transcriptional map of malignant cells from progressing Vκ\*MYC mice. a** Schematic of Vκ\*MYC mouse cohort and experimental workflow for collection of single cells. Graphics created in part using BioRender.com. **b** Disease burden across cohort as determined by M-protein measurements from SPEP. Statistical comparison of multiple groups was performed using a Wilcoxon rank-sum test (two-sided) corrected for multiple testing (Benjamini−Hochberg). Boxplots represent the distribution of each measurement within defined groups, where the central rectangle spans the interquartile range, the central line represents the median, and "whiskers" above and below the box show the value 1.5× the interquartile range. Only P values for statistically significant comparisons are listed. Data points represent measurements from biologically-independent animals (Control ($n = 3$), early-MM ($n = 5$), int-MM ($n = 3$), and active-MM ($n = 7$)). **c** tSNE visualization of 17,504 B lineage cells, coloured by sample ID. **d** tSNE visualization of B lineage cells, coloured by plasma cell gene signature score (Hay SB et al.[28]). **e** tSNE visualization of B lineage cells, coloured by the relative expression of indicated cell-type-specific genes. **f** tSNE visualization of plasma cells identified by criteria displayed in (**d**) and (**e**) coloured by the relative expression of Vκ\*MYC transgene (tMYC). **g** Bar plot showing the distribution of normal vs. malignant plasma cells across samples No malignant cells were detected in EMM2 and thus it was removed from downstream malignant cell analyses. Source data are provided in SourceData_Fig. 1.xlsx. EMM: early-MM, IMM: intermediate-MM, AMM: active-MM.

Previous bulk gene expression studies in MM employ cell selection methods that do not discriminate between normal and malignant plasma cells, thus resulting in potentially contaminated malignant cell expression profiles. We were able to make this distinction in our scRNA-seq data set by measuring Vκ\*MYC transgene (tMYC) expression levels, which revealed a small population of normal plasma cells with significantly lower tMYC expression (Fig. 1f). The expression profiles of this population also scored lower for gene sets comprised of MYC transcriptional targets (Chesi et al.[17], Schuhmacher et al.[29], Menssen et al.[30], Supplementary Fig. 1e–g) further supporting their identity as normal, non-malignant plasma cells. Consistent with this, only normal plasma cells were identified in age-matched control mice, while the proportion of normal plasma cells progressively decreased from early/int-MM to active-MM (Fig. 1g). Moreover, the proportion of malignant cells in each tumour from our scRNA-seq data correlated strongly with initial M-protein measurements ($R = 0.890$, $P = 7.401e−07$, Supplementary Fig. 1h), supporting our use of this marker to define disease progression. Consistent with disease stage classifications defined above, mice in the active-MM group had significantly more malignant cells compared to all other disease groups and control mice (Supplementary Fig. 1i). In contrast, all early-MM and int-MM mice had less than 5% malignant cells (no significant difference across groups). Thus, measuring gene expression profiles at the single-cell level revealed that normal and malignant

plasma cells have distinct transcriptional programs and enabled discrimination of these populations within samples.

**A core malignant program spans the myeloma disease spectrum.** Using the scRNA-seq data from purely malignant cells, we next defined a core molecular program shared by all malignant cells in the Vκ\*MYC mouse model, regardless of disease stage. In this and all subsequent analyses, malignant cells were defined as plasma cells that were not members of the normal plasma cell cluster described above. We first performed differential expression (DE) analysis between the malignant cells at each disease stage and normal plasma cells (Supplementary Data 2). We then selected differentially expressed genes that overlapped in all disease stage groups, which produced a core set of 226 differentially expressed genes (171 upregulated, 55 downregulated) shared by malignant cells at all disease stages (Fig. 2a, Supplementary Fig. 2a, and Supplementary Data 3). These genes included tMYC and Ccnd2, consistent with the hypothesis of early and universal cyclin D dysregulation in the pathogenesis of MM[16]. To clarify the biological relevance of the core malignant cell program, we performed gene set enrichment analysis using the 226 shared differentially expressed genes and plotted the top 20 positively and negatively enriched terms (Fig. 2b and Supplementary Data 4). As expected, positively enriched terms were associated with MYC transcriptional regulation, myeloma pathogenesis, and

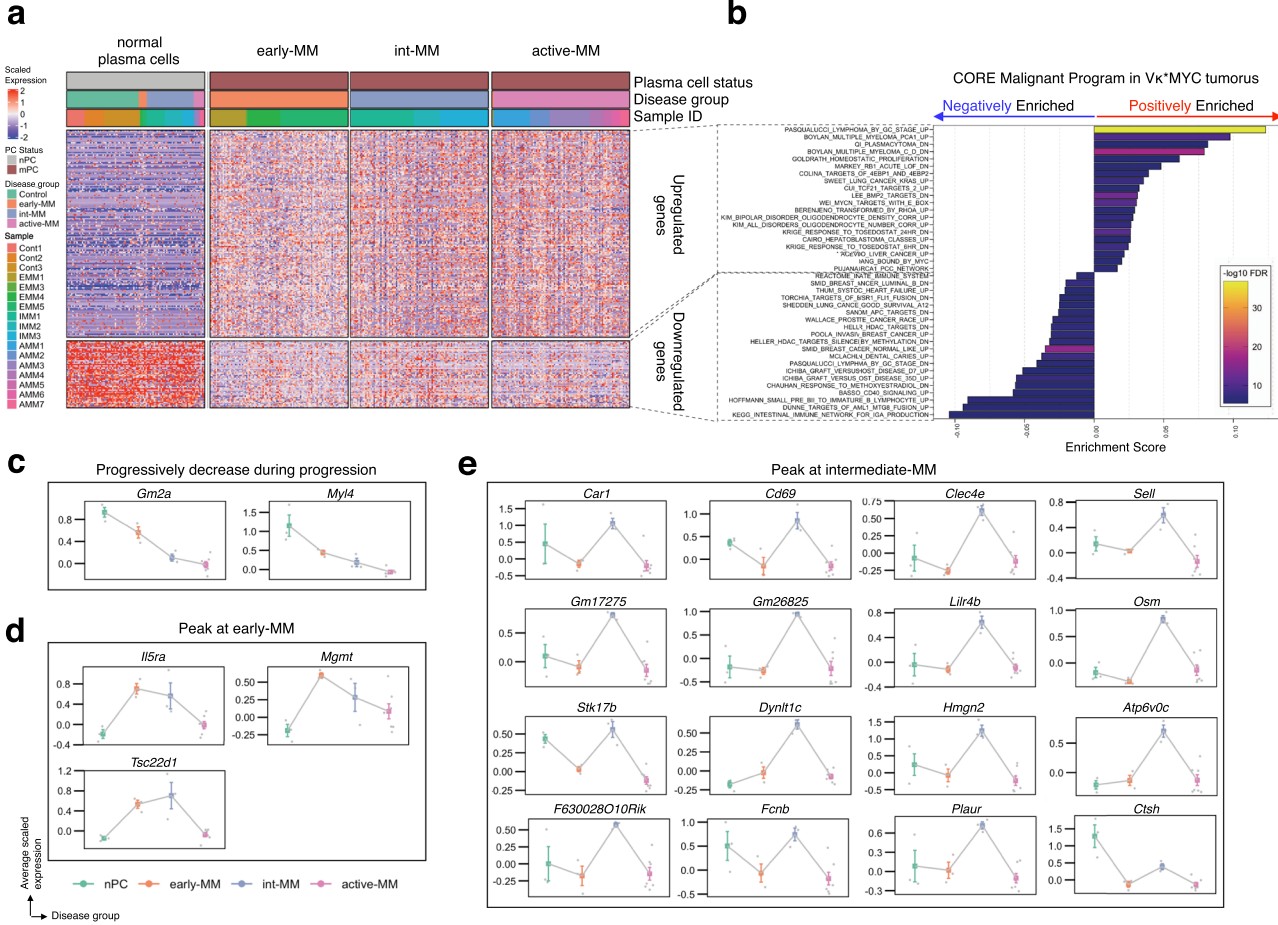

**Fig. 2 Core versus disease-stage specific gene expression programs in malignant cells from Vκ*MYC mice. a** Heatmap of differentially expressed genes shared by all malignant cells in Vκ*MYC mice compared to normal plasma cells (FDR < 0.05). Heatmap is split vertically to show normal plasma cells (nPC) versus malignant plasma cells (mPC), the latter of which is further split by disease stage group. The upper and lower panels of the heatmap separate upregulated and downregulated genes, respectively. A subset of 100 randomly selected cells per disease stage group are shown and data represent scaled expression values (any values outside a range of −2 to 2 were clipped). **b** Top 20 positively/negatively enriched terms from MSigDB gene set enrichment analysis (H, C2, C6, FDR < 0.05) computed using core upregulated/downregulated genes identified by DE analysis in (**a**). **c–e** Disease stage-specific genes that are significantly differentially expressed between disease stage groups. Coloured dots represent the mean expression of disease stage samples for each gene, with error bars depicting the standard error of the mean. Statistical comparisons were performed using a two-sided t-test with subsequent correction for multiple testing (Bonferroni). Grey data points represent mean expression of respective genes in cells from each biologically-independent animal (Cont1 = 44 cells, Cont2 = 72 cells, Cont3 = 148 cells, EMM1 = 45 cells, EMM4 = 52 cells, EMM5 = 71 cells, IMM1 = 206 cells, IMM2 = 88 cells, IMM3 = 149 cells, AMM1 = 2,003 cells, AMM2 = 830 cells, AMM3 = 1,379 cells, AMM4 = 822 cells, AMM5 = 302 cells, AMM6 = 323 cells, AMM7 = 310 cells). Genes are grouped according to the pattern of expression throughout progression. Source data are provided in SourceData_Fig. 2.xlsx.

general cancer biology. Interestingly, many of the negatively enriched terms from this analysis were associated with immune processes such as graft-versus-host disease and innate immunity, suggesting that malignant cells express a molecular program that promotes immune dysregulation. Indeed, this would be consistent with results from Zavidij et al. who demonstrated that immune dysfunction occurs early in myeloma pathogenesis[13]. Other signatures were also identified in this analysis that implicates previously unreported processes in Vκ*MYC disease biology including signatures related to knockdown of the tumour suppressor *TCF21*, *KRAS* mutations, and constitutive activation of the *RHOA* oncogene. This may in turn suggest that additional oncogenic hits are acquired early in the course of Vκ*MYC tumourigenesis and persist throughout disease evolution.

**Subtly distinct expression programs underpin progression.** The analysis above revealed a set of overlapping genes shared by

malignant cells across the disease spectrum, so we next asked whether distinct molecular programs emerge longitudinally throughout progression. By employing DE analysis, we defined the temporal expression patterns that are specific to malignant cells from each stage of progression (see "Methods", Fig. 2c–e, Supplementary Fig. 2b, c, and Supplementary Data 5). This revealed 21 genes with expression levels that changed significantly throughout progression (Fig. 2c–e) and whose longitudinal pattern of expression coincided with one of three different groups. The first group of genes consisted of *Gm2a* and *Myl4* whose expression progressively decreased during progression (Fig. 2c). The second group, whose expression peaked at the early-MM disease stage (Fig. 2d), consisted of *Il5ra*, a cytokine receptor, *Mgmt*, a methyltransferase crucial for genome stability, and *Tsc22d1*, a pro-apoptotic tumour suppressor. This, therefore, suggests that malignant cells at early disease stages increase expression of apoptotic regulators and genome stabilizers but that these processes may be progressively lost during progression. The

third group of genes demonstrated a temporal pattern characterized by peak expression in the intermediate stage of disease ("Peak-Intermediate", Fig. 2e) and included genes related to immune processes such as cytokine signalling (*Osm*, *Lilr4b*), the complement system (*Fcnb*), and cell adhesion (*Clec4e*, *Sell*). Genes whose expression peaked during the intermediate stage of progression also included *F630028O10Rik*, a lncRNA reported to play a role in modulating tumour angiogenesis[31], *Plaur*, a urokinase receptor involved in cell migration, cell cycle regulation, and cell adhesion[32], and *Ctsh*, a lysosomal cysteine proteinase whose expression has been correlated with malignant progression of prostate tumours[33]. Thus, disease progression in Vκ*MYC mice is accompanied by transcriptional variability within the malignant cell compartment. However, given how few differentially expressed genes were shared by mice within a given disease stage group, we hypothesized that intra/intertumoural heterogeneity may be present in Vκ*MYC mice and account for the subtlety of expression changes observed longitudinally.

**CNVs are a source of tumour heterogeneity in Vκ*MYC mice**. To evaluate whether malignant cells demonstrate heterogeneity driven by somatic genome alterations, we inferred CNVs from our scRNA-seq data using the InferCNV[34] algorithm (see "Methods" for details). This analysis revealed extensive copy number variation in the malignant compartment of all Vκ*MYC mice (Fig. 3a and Supplementary Fig. 3a). This included previously reported CNVs such as gain of chr1, chr6, chr16, chr18, and loss of chr5 and chr14[35]. CNVs not previously reported in this model included subchromosomal losses in chr3, chr9, and chr17 and subchromosomal gains in chr15. Notably, many of these CNVs were not shared by all Vκ*MYC mice, even within mice from the same disease stage (e.g., loss of chr12 in AMM1, gain of chr18 in IMM3). By partitioning cells into groups having consistent patterns of CNVs, we observed that Vκ*MYC tumours are comprised of 4–8 malignant cell subpopulations with distinct CNV profiles (Fig. 3a and Supplementary Fig. 3a, b), underscoring the power of scRNA-seq to delineate malignant subpopulations not detected by bulk approaches. The presence of multiple cell subpopulations with distinct CNV profiles was evident at all disease stages (Fig. 3a and Supplementary Fig. 3a, b). This, therefore, suggests that genomic diversification of tumours begins early in the evolution of myeloma, similar to what has been described in humans[9,36], and continues throughout progression.

The above analysis supports that the malignant cell compartment of Vκ*MYC mice exhibits intra and intertumoural heterogeneity, despite being driven by the same oncogenic MYC transgene. However, it also reveals several CNVs that are shared by subpopulations of malignant cells throughout the disease spectrum, most notably, loss of chr5. To better understand the biological underpinnings of chr5 loss in Vκ*MYC mice, we performed enrichment analysis using genes differentially expressed between cells with chr5 deletion (del(5)) and cells with wild-type chr5 (chr5WT) (Supplementary Data 6, 7). Not surprisingly, this analysis revealed activation of processes related to chromosomal instability in cells with del(5) including "DNA double-strand break response", "Recruitment of ATM-mediated phosphorylation of repair and signalling proteins at DNA double-strand breaks", "Processing of DNA double-strand break ends" and "DNA Damage/Telomere Stress Induced Senescence" (Supplementary Fig. 3c). To gain insight into how the loss of chr5 may translate to human myeloma patients, we performed mouse-to-human mapping of orthologous genes (Supplementary Fig. 3d and Supplementary Data 8). The vast majority of genes on mouse chr5 mapped to three human chromosomes: chr4, chr7,

and chr12 (930/1196 genes). However, a subset of genes also mapped to human chromosomes reportedly deleted in precursor myeloma patients (chr1p, chr2q, and chr13q)[37], suggesting that these genes may mediate the pathogenic effects of chr5 loss in Vκ*MYC mice and in turn play a role in myeloma progression from precursor disease.

Inferred CNV analysis of our scRNA-seq data supports the existence of intratumoural heterogeneity, which is a well-established phenomenon in cancer. However, whether these events drive transcriptional heterogeneity and the resulting biological effects are less understood. Therefore, we mapped CNV-defined subpopulations to transcriptional clusters (Fig. 3b) determined by unsupervised clustering of scRNA-seq data from Vκ*MYC mice with active-MM. Similar to our CNV analysis, transcriptional clustering revealed that the malignant cell compartment of Vκ*MYC mice is comprised of multiple molecular subpopulations per mouse (3–13 transcriptional clusters, Fig. 3b). The fact that samples with a similar number of cells did not have the same number of transcriptional clusters (e.g., AMM2 and AMM4, AMM6 and AMM7) supports that the range in the number of transcriptional clusters is not just an artifact of differences in the number of cells profiled across samples. We then evaluated whether this transcriptional heterogeneity was driven by the subclonal CNVs inferred above by exploring the distribution of CNV-driven subpopulations within each transcriptional cluster. In doing so, we observed instances of majority CNV-driven transcriptional clusters (Fig. 3c, d). For example, in AMM1, transcriptional cluster 4 was largely comprised of cells from CNV subpopulation 3, defined by del(5) and del(12). Similarly, in AMM4, transcriptional cluster 0 was largely comprised of cells from CNV subpopulation 1, defined by subchromosomal gain of chromosome 9. This supports that subclonal CNVs can have a significant effect on the formation of distinct transcriptional clusters. However, the majority of CNV-driven subpopulations were distributed across several transcriptional clusters (Fig. 3d) and we did not find a significant correlation between the number of CNV-driven subpopulations and the number of transcriptional clusters ($R = -0.597$, $P = 0.1567$, Supplementary Fig. 3e). Moreover, this transcriptional heterogeneity was retained when potentially confounding genes associated with dissociation[38] and mitochondrial/ribosomal/cell cycle genes were removed (Supplementary Fig. 4). Thus, our data robustly support that transcriptional variability must be driven by additional sources beyond subclonal CNV events. In turn, we next focused on defining the biological pathways associated with transcriptional clusters in mice with active-MM.

**Conserved transcriptional programs in Vκ*MYC tumours**. To further explore drivers of heterogeneity in the transcriptional clusters defined in Fig. 3b, we combined DE and enrichment analysis to define cluster-specific pathways within Vκ*MYC tumours (Supplementary Data 9, 10). Significantly enriched pathway terms for each transcriptional cluster were then compared pairwise to all other cluster-specific pathway terms using a Jaccard Similarity Index, which revealed two distinct transcriptional programs with representative clusters from all seven active-MM mice (Fig. 4a): Similarity Program A (11 clusters, mean Jaccard Index = 0.641) and Similarity Program B (9 clusters, mean Jaccard Index = 0.660). The remaining 21 clusters displayed limited similarity (mean Jaccard Index = 0.054), supporting that transcriptional divergence is a characteristic feature of disease progression. As depicted in Supplementary Fig. 5, the molecular processes associated with these divergent clusters included the immune system (innate immunity, cytokine

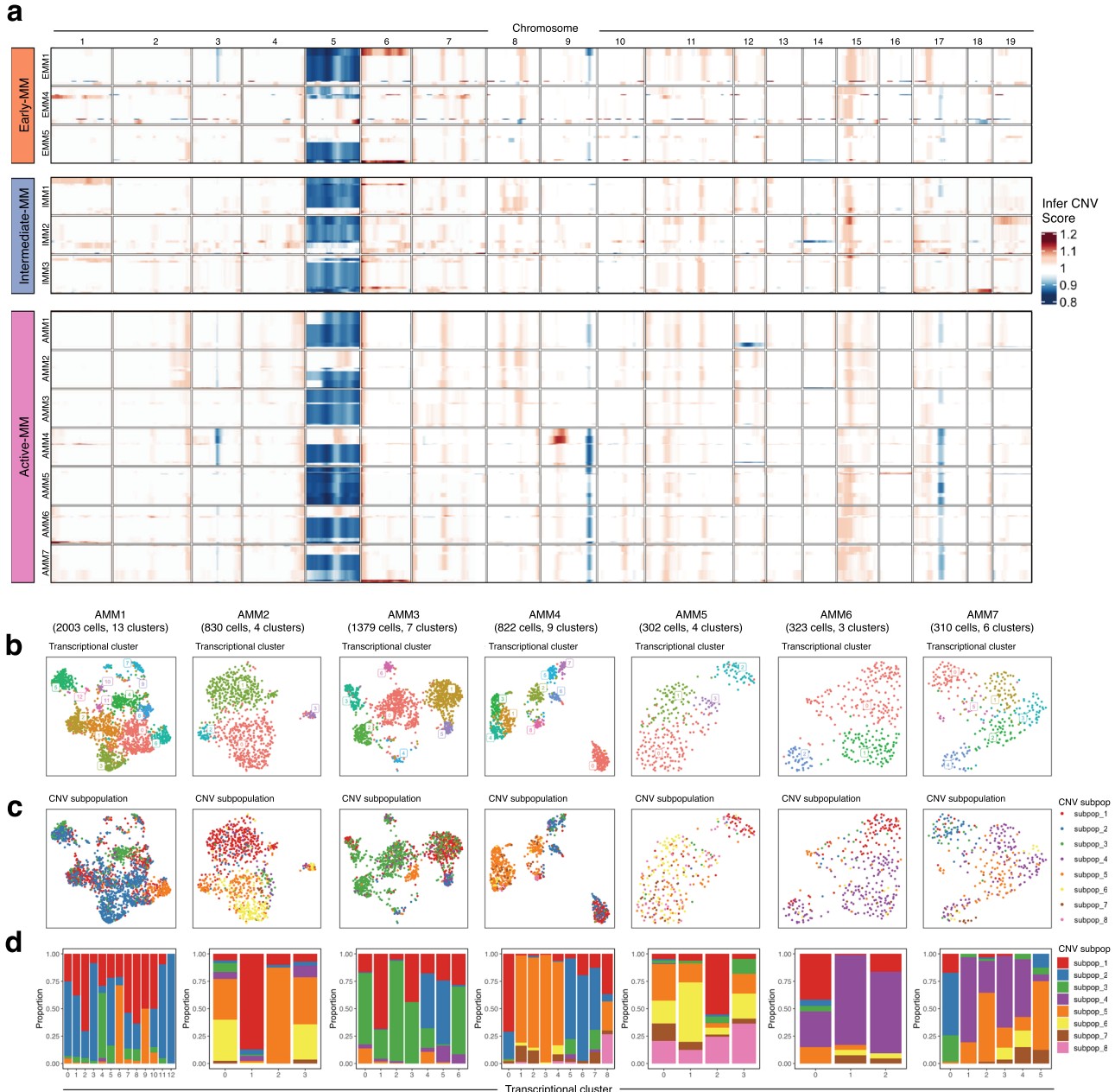

**Fig. 3 Drivers of intratumoural heterogeneity in the malignant cell compartment of Vκ*MYC mice. a** Heatmap of genome-wide copy number variations (CNVs) inferred from scRNA-seq data of malignant plasma cells as determined using InferCNV[34]. Columns represent genome position across chromosomes. Rows represent CNVs averaged by intra-sample CNV subpopulation, which were identified using the ward.D2 hierarchical clustering/random forest method implemented by analysis_mode = 'subclusters' in inferCNV. CNV-driven subpopulation sizes ranged from 1 to 979 cells (median 44). The height of each CNV subpopulation is proportionate to its fractional composition within a given tumour. **b** UMAP visualization of malignant cells from each active-MM mouse coloured by transcriptional cluster. Gene expression-driven cluster sizes ranged from 10 to 474 cells (median 79). **c** UMAP visualization of malignant cells from each active-MM mouse coloured by CNV subpopulation. **d** Bar plot showing the distribution of CNV subpopulations (fill) across transcriptional clusters (x-axis). Results for (**b–d**) are organized for each active-MM mouse in columns, with subject names and the number of cells/transcriptional clusters listed above. Source data are provided in SourceData_Fig. 3.xlsx. EMM: early-MM, IMM: intermediate-MM, AMM: active-MM.

signalling), cellular responses to stress (heat stress, metal ions), and signal transduction through mediators such as MAP kinase, nuclear receptors, and WNT. Thus, in addition to CNV-level events, transcriptional heterogeneity in the malignant compartment of Vκ*MYC mice is also driven by other mechanisms, which likely include external contributions from the tumour microenvironment.

Although myeloma tumours undergo transcriptional divergence during progression, certain biological processes were

shared by a subpopulation of cells common to all tumours. We, therefore, sought to investigate the biological processes driving these shared transcriptional programs. Previous single-cell cancer studies have identified a recurrent cell proliferation-related transcriptional program in many human cancers[39] and thus we explored whether either of the shared transcriptional programs were associated with this process. Indeed, Similarity Program B clusters were enriched for pathway terms related to the cell cycle including "Chromosome Maintenance", "Cell Cycle Checkpoints"

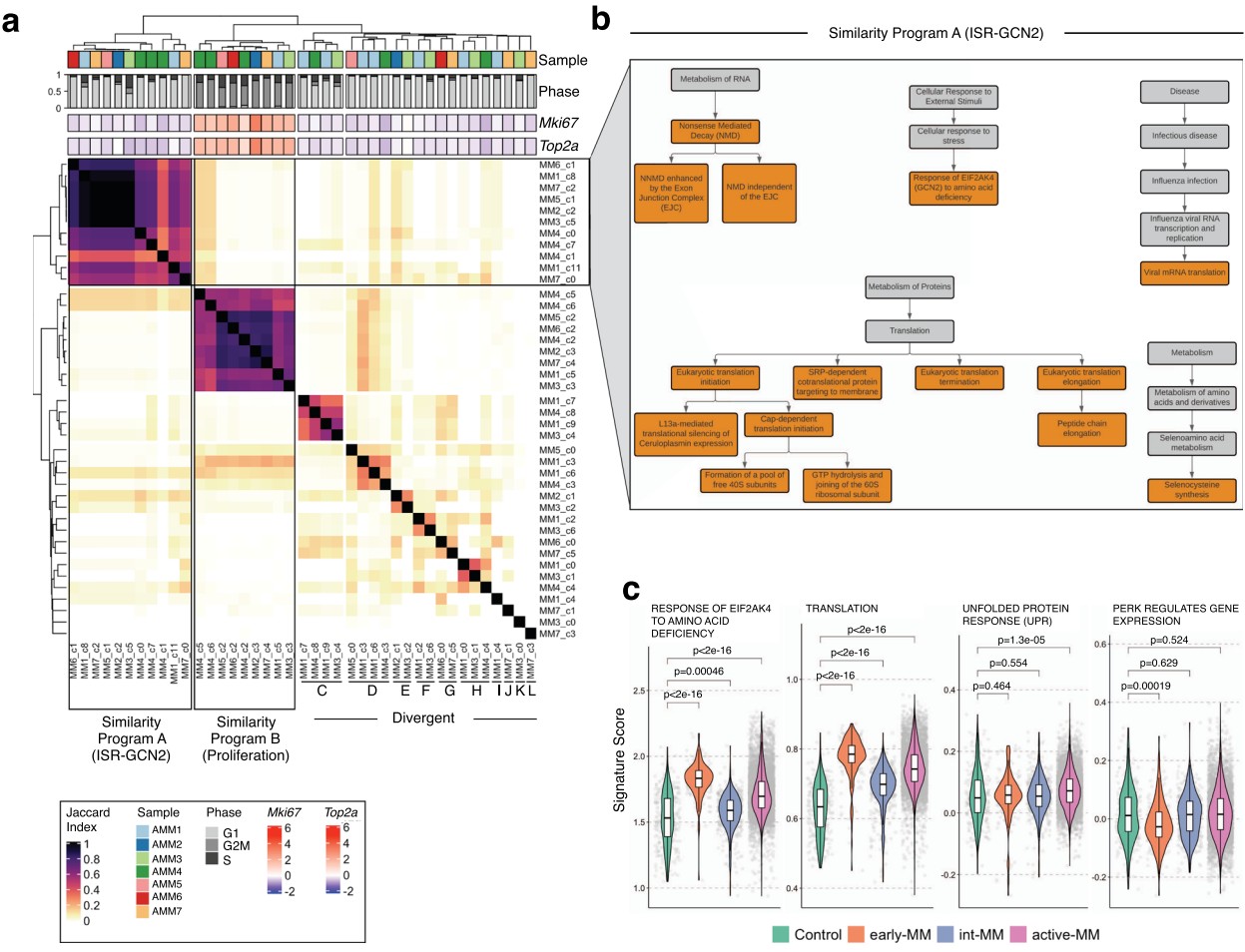

**Fig. 4 Molecular programs driving intratumoural heterogeneity in Vκ*MYC mice with active-MM. a** Heatmap of Jaccard Index between significantly enriched Reactome terms across 41 intra-tumour malignant cell clusters. Groupings of clusters with increased similarity (Similarity Programs) were determined according to the complete linkage method for hierarchical clustering and are labelled below the heatmap. Columns are annotated with information related to sample identity, cell cycle phase, and *Mki67/Top2a* expression. **b** Map of Reactome terms with significant enrichment in malignant cell clusters from Similarity Program A (ISR-GCN2). The full hierarchy of each Reactome pathway is shown for context but only significantly enriched shared pathways are highlighted in orange. **c** Gene set scoring for indicated Reactome signatures calculated using Seurat's AddModuleScore across disease groups in Vκ*MYC data (Control = 264 cells, early-MM = 168 cells, int-MM = 443 cells, active-MM = 5,969 cells). Boxplots within violin plots represent the distribution of each measurement within defined groups, where the central rectangle spans the interquartile range, the central line represents the median, and "whiskers" above and below the box show the value 1.5× the interquartile range. Statistical comparison of multiple groups (normal PCs vs. each Vκ*MYC disease group) was performed using the Wilcoxon rank-sum test (two-sided) corrected for multiple testing (Benjamini−Hochberg). Source data are provided in SourceData_Fig. 4.xlsx.

and "Cell Cycle, Mitotic" (Supplementary Data 10), expressed high levels of canonical cell cycle genes such as *Mki67* and *Top2a* (Fig. 4a), and were predominantly comprised of cycling cells (Fig. 4a, mean G2/M 73.99 ± 11.16% in Similarity Program B clusters vs. 2.95 ± 4.28% for all other clusters). This is consistent with the subpopulation of highly proliferative malignant cells reported in MM patients, which typically constitutes ~2% of the tumour[40]. Although this population is more prevalent in Vκ*MYC mice (mean 12.9 ± 7.7%, Supplementary Fig. 6a), similar ranges have been observed in MM patients, with higher proliferative indexes correlating to worse outcomes[41]. These results may therefore reflect the adverse effects of MYC dysregulation in myeloma biology. Finally, these proliferative clusters were comprised of multiple CNV subclones suggesting that proliferative programs have a stronger effect on gene expression than subclonal CNV events (Supplementary Fig. 6b).

We next investigated the other set of terms shared by a subpopulation of cells in all active-MM tumours (Similarity Program A), which revealed an interesting pattern of biological

pathways including "Translation" and "Response of *EIF2AK4* to amino acid deficiency" (Fig. 4b and Supplementary Data 10). GCN2 (the protein encoded by *EIF2AK4*) represents one arm of the Integrated Stress Response (ISR) pathway, whereby cells adapt to various stresses such as viral infection, heme deprivation, unfolded protein response (UPR) and, in the case of GCN2, amino acid deprivation[42] (Supplementary Fig. 6c). Activation of the ISR pathway, in turn, promotes cellular adaptation to overcome these stress conditions. Thus, activation of the ISR-GCN2 pathway may represent a stress response pathway and protective mechanism employed by a subset of malignant cells in MM. Notably, a similar set of terms related to ISR-GCN2 was also detected using non-negative matrix factorization (Supplementary Fig. 6d), as were a separate set of terms related to proliferative processes akin to Similarity Program B described above (Supplementary Fig. 6e), further supporting the existence of these subclonal malignant programs in Vκ*MYC tumours.

To understand how the ISR-GCN2 pathway relates to disease progression, we quantified activation of this pathway in

malignant cells from all Vκ*MYC mice using the "Response of *EIF2AK4* to amino acid deficiency" gene set (ISR-GCN2 signature, $n = 87$ genes, Supplementary Fig. 6f, Supplementary Data 11). Not only was the ISR-GCN2 pathway upregulated in malignant cells from all disease groups compared to normal plasma cells, but the highest level of ISR-GCN2 activation was also observed in malignant cells from mice with the earliest stages of disease (early-MM) (Fig. 4c). Notably, activation of ISR-GCN2 was not positively associated with the UPR pathway, nor did it positively correlate with other arms of the ISR pathway such as PERK (Fig. 4c and Supplementary Fig. 6g). However, ISR-GCN2 activity was strongly correlated with protein translation ($R = 0.847$, $P = 2.2e-16$, Fig. 4c, Supplementary Fig. 6g), supporting that activation of ISR-GCN2 and protein translation are related. Given the role of MYC in promoting protein synthesis, these results put forth a model of disease in Vκ*MYC mice whereby ISR-GCN2 activation occurs early in disease pathogenesis to tolerate cellular stress caused by MYC activation and the resulting excessive protein translation (Supplementary Fig. 6c). However, since MYC remains activated throughout Vκ*MYC progression, the subsequent decrease in ISR-GCN2 activity in int-MM mice (Fig. 4c) and the subclonal nature of ISR-GCN2 activation in active-MM mice (Fig. 4a) suggests that GCN2 activation may be regulated by other mechanisms that drive amino acid deficiency. In myeloma, we speculate that this could be caused by excess production of immunoglobulin proteins, which is a hallmark physiological process in myeloma.

The observation that ISR-GCN2 activation is subclonal in active-MM mice, but has the highest activity in early-MM mice suggests that molecular variability in advanced disease may reflect residual programs from precursor disease. To support this model of progression, we scored the transcriptional clusters in active-MM mice using highly expressed genes in int-MM mice derived from our longitudinal analysis ("Peak-Intermediate", Fig. 2e). In doing so, we found that "Peak-Intermediate" genes were highly expressed in only a subset of cells from active-MM mice (Supplementary Fig. 6h) and that the pathways associated with these clusters were also related to immune processes (Supplementary Fig. 5, Program C). Thus, these data suggest that gene expression programs that once defined the landscape of malignant cells in precursor disease stages become subclonal in advanced disease as tumours diversify (Fig. 5).

**GCN2 inhibition represents an anti-MM therapeutic strategy.** Since ISR-GCN2-driven subclones were detected in all mice with active-MM, we next evaluated whether this pathway has relevance in human myeloma disease. To do this, we scored publicly-available expression data from primary BM plasma cell samples using the ISR-GCN2 gene signature. This included bulk expression data from Chng et al.[43] (134 samples: 15 healthy donor, 22 MGUS, 24 SMM, 73 newly diagnosed MM) and a scRNA-seq data set from Ledergor et al.[12] (35 samples: 11 healthy donors, 6 MGUS, 6 SMM, and 12 MM). In both data sets, a significant progressive increase in ISR-GCN2 scores was detected across disease stages from MGUS to MM (Fig. 6a, b), suggesting that activation of this transcriptional program is associated with progression from precursor disease to overt MM. Considering each patient individually, our analysis also supports that the ISR-GCN2 pathway is more highly active in a subset of patients with MM (Supplementary Fig. 7a, b). Similar to Vκ*MYC mice, activation of ISR-GCN2 was strongly associated with protein translation in both Chng et al. and Ledergor et al. data sets (Supplementary Fig. 7c, d), but not with UPR or PERK-related ISR (Supplementary Fig. 7c, d). Taken together, these results

confirm that the ISR-GCN2 pathway is highly active in a subset of myeloma patients, correlates with increased levels of protein translation, and is associated with disease progression. Thus, we next assessed whether this pathway is necessary for myeloma cell survival.

GCN2 has recently emerged as a promising drug target in solid tumours and haematological malignancies[43–47], but it is unknown whether this pathway represents a therapeutic vulnerability in MM. We, therefore, treated a panel of human myeloma cell lines (HMCLs, $n = 9$) with the small molecule GCN2 inhibitor, GCN2iB[44], to assess whether myeloma cells rely on the ISR-GCN2 pathway for survival. This analysis revealed that treatment with GCN2iB has anti-myeloma activity, but induced a range of responses in HMCLs (Fig. 6c), which we hypothesized may correlate with differences in basal ISR-GCN2 pathway activation. Indeed, by scoring publicly-available HMCLs expression data using the ISR-GCN2 signature, we found a statistically significant inverse association between ISR-GCN2 score and viability after treatment with GCN2iB ($R = -0.605$, $P = 0.028$, Fig. 6d), supporting that cell lines with higher ISR-GCN2 activity are more sensitive to GCN2iB. Notably, MYC signature scores determined for each HMCL did not correlate with GCN2iB response ($R = -0.067$, $P = 0.829$, Supplementary Fig. 8a) or ISR-GCN2 activity ($R = 0.077$, $P = 0.802$, Supplementary Fig. 8b), suggesting that ISR-GCN2 activation may not be uniquely related to MYC dysregulation in human myeloma. Nonetheless, in further support of a role for the ISR-GCN2 pathway in myeloma survival, GCN2iB-sensitive HMCLs, as well as Vκ*MYC tumours (Vκ12598), undergo apoptosis upon treatment with GCN2iB (Fig. 6e, f, and Supplementary Fig. 8c, d). Consistent with these effects, genetic knockout of GCN2 in GCN2iB-sensitive HMCLs, RPMI-8226 and OPM2, reduced cell viability (Fig. 6g and Supplementary Fig. 8e), but had no effect on the viability of GCN2iB-insensitive HMCLs, MM1S, and XG7 (Supplementary Fig. 8e, f). Thus, our data support that cellular responses to stress via the ISR-GCN2 pathway are active in a subset of HMCLs, and in turn, may represent an important process driving the progression of plasma cell neoplasms. These findings not only provide an anti-myeloma target, but also an accompanying biomarker that can guide personalized therapeutic utility.

## Discussion

Recent single-cell[12,13] and large-scale genomic[6,48–50] studies profiling the spectrum of myeloma progression have revealed a highly complex molecular landscape within and between patients. This in turn has made defining mechanisms by which myeloma evolves from a precursor stage to active disease challenging. In the current study, we extensively characterized the molecular composition of the malignant cell compartment during disease evolution by profiling the inferred CNV status and cellular states of individual cells in the Vκ*MYC mouse model. This allowed for a controlled, systematic comparison of gene expression programs across disease stages in a model that is highly faithful to human disease and is universally driven by a homogeneous transforming event. Using a longitudinal single-cell transcriptional analysis across the disease progression spectrum, we revealed patterns of transcriptional divergence at distinct disease stages and identified overlapping subpopulation-level gene expression programs. We also demonstrated the clinical relevance in myeloma patients of a shared transcriptional program identified from analysis of Vκ*MYC tumours using publicly available human single-cell and bulk gene expression data sets and showed that targeting this program has anti-MM activity.

Although tumour heterogeneity is well established in myeloma, our integrated analysis of single-cell CNVs and transcriptional

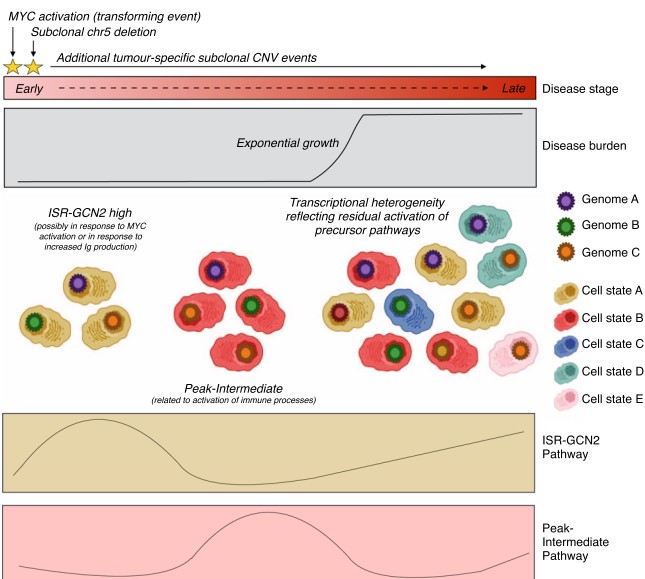

**Fig. 5 Proposed model for patterns of tumour heterogeneity during myeloma disease progression in Vκ*MYC mouse model.** Graphics created in part using BioRender.com.

profiles demonstrates further complexity that is likely to drive progression and mechanisms of resistance. Indeed, even in a model where host, environment, and initial oncogenic driver are homogeneous, we unexpectedly observed extensive molecular and phenotypic intra and intertumoural heterogeneity. To better understand if and how these features contribute to differences in disease trajectories and rates of progression, future studies could consider exploring molecular changes that influence disease progression within an individual mouse via serial sampling. This approach could also enable the characterization of continuous changes such as cancer cell plasticity that may contribute to intratumoural heterogeneity and could further remove biological variability between mice that may have impacted our DE analysis across disease stages. Nonetheless, we found that Vκ*MYC tumours are comprised of 4–8 distinct subpopulations of malignant cells when defined using inferred CNV profiles, which is consistent with previous whole-exome sequencing and single-cell genetic analyses that uncover evidence for 2–6 subclones at myeloma diagnosis[51]. Interestingly, when we defined subclones in Vκ*MYC mice using transcriptional variability, we observed more substantial intratumoural heterogeneity (3–13 clusters) compared to when defined by CNV profile. We acknowledge the possibility that these differences may be a result of different clustering algorithms used to group the data. However, it is notable that our finding of lack of overlap between some

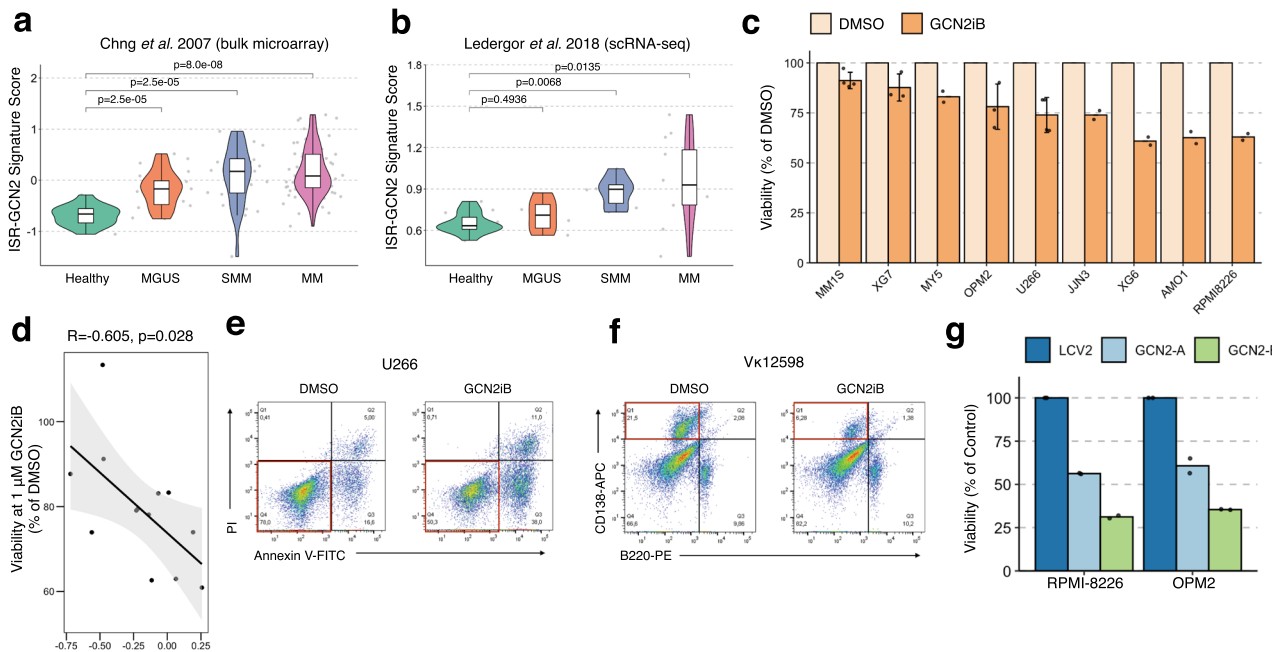

**Fig. 6 ISR-GCN2 pathway is progressively activated in myeloma patients and supports myeloma cell survival. a** ISR-GCN2 gene signature scoring in publicly-available microarray patient data from Chng et al.[43] calculated by taking the mean of scaled expression values for genes from the ISR-GCN2 gene set. **b** ISR-GCN2 gene signature scoring calculated using Seurat's AddModuleScore in publicly-available scRNA-seq patient data from Ledergor et al.[12] across disease groups. Statistical comparisons of multiple groups (Healthy vs. each disease group) in (**a**) and (**b**) were performed using the Wilcoxon rank-sum test (two-sided) corrected for multiple testing (Benjamini−Hochberg). Boxplots in (**a**) and (**b**) represent the distribution of each measurement within defined groups, where the central rectangle spans the interquartile range, the central line represents the median, and "whiskers" above and below the box show the value 1.5× the interquartile range. **c** MTT analysis of GCN2iB treatment in HMCLs (1 μM, 72 h). Data represent the mean of biological replicates ($n = 2$ for MY5, JJN3, XG6, AMO1, and RPMI8226; $n = 3$ for XG7, OPM2; $n = 4$ for MM1S, U266) with error bars representing standard deviation shown for samples with more than two biological replicates. **d** Pearson correlation (cor.test, two-sided) between the ISR-GCN2 gene signature score (x-axis) and viability relative to DMSO (1 μM GCN2iB, y-axis). The linear regression line is plotted in black with confidence interval shaded grey. Each dot represents one HMCL. **e** Flow cytometric analysis of apoptosis in U226 cells after 48 h treatment with 5 μM GCN2iB. **f** Flow cytometric analysis of apoptosis in Vκ12598 tumour cells (CD138+/B220−) after 48 h treatment with 5 μM GCN2iB. Scatter plots in (**e**) and (**f**) are representative of multiple independent experiments (see Supplementary Fig. 8c, d). Quadrants corresponding to viable tumour cells are highlighted in red. **g** Bar plots depicting cell survival, as determined by trypan blue assay, in GCN2 knockout HMCLs. Bar heights represent mean relative viability from two independent experiments (as shown by individual data points). Source data are provided in SourceData_Fig. 6.xlsx.

CNV-level subpopulations and transcriptionally-defined subclones is consistent with the recent findings from Fan et al.[52]. This, in turn, supports that genetic events are not the sole drivers of divergent malignant programs and that molecular heterogeneity is likely driven by additional non-genetic factors such as epigenetic regulators or differences in the tumour microenvironment. Thus, the results demonstrate the utility of our approach for unravelling the complexity of myeloma tumours that interplay in cancer progression.

In this study, we attempted to define longitudinal changes in the molecular programs of malignant cells, however, we found very few genes that were differentially expressed between each stage of disease progression. It is possible that minor differences in disease burden between early-MM and int-MM and/or biological variability between mice may account for the lack of broad differences in gene expression between these disease stage groups. However, our observations are consistent with those reported by Storti et al. who performed bulk microarray profiling of CD138-selected cells from patient BM and similarly found that the transcriptomic profile of myeloma cells remains substantially unchanged throughout progression[10]. We can further conclude that the lack of change in gene expression profiles is not due to contaminating normal plasma cells that may have been present in the Storti et al. study, since we were able to remove these cells from our data in silico. Rather, we suspect that failure to detect substantial transcriptional changes between disease stages is the result of intra and intertumour heterogeneity. Indeed, we show that the malignant cell compartment is comprised of multiple CNV-driven and transcriptional subpopulations, even in mice where disease is not yet detected by conventional methods. Thus, it is possible that genetic changes associated with disease progression lie in distinct tumour subclones that may be obscured in bulk sequencing analysis. It would follow therefore that deciphering predictive biomarkers of progression will require large-scale single-cell studies of progressing and non-progressing precursor patients.

We acknowledge that the small cell numbers profiled from mice with earlier disease confounds our estimation of intratumoural heterogeneity at these time points. This limitation can and should be addressed in future studies by experimentally enriching for malignant cells prior to scRNA-seq profiling. Nonetheless, our inferred CNV analysis supports the presence of distinct malignant cell populations at all stages of disease and further shows that these subclones persist throughout disease progression. This pattern of progression is consistent with the "static model" recently proposed by Bolli et al.[6] In this model, subclonal architecture is retained as disease progresses from SMM to MM, such that progression reflects the time it takes to accumulate sufficient disease burden, rather than the acquisition of an additional oncogenic hit. Thus, findings from our study may be most applicable to tumours from SMM patients that follow this evolutionary path to progression. The persistence of certain subclonal CNVs, such as loss of chr5, throughout progression supports the notion that disease evolution in the Vκ*MYC model is shaped by positive selection, as recently suggested by Diamond et al.[53] However, our findings add an additional layer to these models whereby transcriptional divergence is a characteristic feature of progression that reflects previously dominant molecular programs that may have been selected against during progression (Fig. 5). This supports the use of preventative treatment strategies in the precursor setting, when tumours are less molecularly complex and potentially easier to target.

Inferred CNV analysis revealed a high prevalence of subclonal chr5 loss, which occurred before the disease could be detected by conventional serum markers. Previous studies have demonstrated the loss of chr5 in the Vκ*MYC model[35,54], but we refine this observation by showing that chr5 loss is an early and subclonal event. It is therefore highly likely that loss of chr5 is a secondary hit that occurs shortly after the initial transforming event of MYC dysregulation (Fig. 5). We also show that loss of chr5 is a highly recurrent event in Vκ*MYC tumours and more common than previously reported by Chesi et al. who found an incidence of approximately 50%[35,54]. The discrepancy in incidence may reflect differences in technologies employed for the analyses (scRNA-seq vs. array-based comparative genomic hybridization on bulk plasma cells, respectively). Alternatively, the universality of chr5 loss reported in our study may be explained by the use of Vκ*MYC mice backcrossed onto C57BL/KaLwRij mice since tumours derived from this strain (5TMM) also demonstrate loss of chr5[35,54,55]. Nonetheless, the recurrent and early nature of chr5 loss in Vκ*MYC mice suggests the presence of one or more driver genes that may be implicated in promoting disease progression. Although we cannot ascertain the human-equivalent of chr5 loss in mice from our analysis, many of the genes on mouse chr5 genes are located on human chromosomes 13, the most commonly reported CNV in precursor disease[37]. Future work should therefore focus on elucidating the corresponding alternation to mouse chr5 in human myeloma patients and further characterizing its role in myeloma biology.

Although Vκ*MYC mice with active-MM demonstrated significant molecular divergence, we identified a subpopulation of cells in every tumour with a shared transcriptional program related to protein translation and the response to amino acid deprivation via the stress-sensing kinase GCN2. We in turn show that inhibition of GCN2 has anti-myeloma activity in a panel of HMCLs, consistent with previous studies that show GCN2 to be critical for cancer cell survival in solid tumours under conditions of nutrient deprivation[44,56]. Our findings lend relevance to the model of myeloma whereby hallmark characteristics of long-lived BM plasma cells like secretory function and protein production are retained during myelomagenesis and in turn, represent highly effective therapeutic vulnerabilities[57]. Although the exact role of GCN2 in myeloma disease progression remains to be established, it is possible that GCN2 activation may represent an adaptive response to amino acid deprivation during increased immunoglobulin production. Alternatively, GCN2 activation may be a downstream consequence of tumour-associated immune cell-induced amino acid shortages in the tumour microenvironment[58,59], which if dissected could reveal combinatorial immune strategies for MM patients. Thus, follow-up studies including those that incorporate tumour-associated immune cells are warranted to delineate the role of GCN2 in myeloma biology.

Given that MYC promotes global protein synthesis and drives malignant transformation in the Vκ*MYC model, it is tempting to speculate that there is a functional interaction between MYC and GCN2 in myeloma. Indeed, previous studies have described MYC-induced activation of the ISR-GCN2 pathway[60]. However, several of our findings support that activation of the ISR-GCN2 pathway is not directly regulated by MYC in the Vκ*MYC model and human myeloma. For instance, constitutive activation of the MYC transgene in Vκ*MYC mice is the transforming event and thus, present in all cells. However, we observed low or subclonal levels of ISR-GCN2 activation in malignant cells from mice with int-MM and active-MM, respectively. Similarly, our in vitro studies revealed that activation of ISR-GCN2 in HMCLs was not correlated with MYC signature scores and that response to GCN2 inhibition was dependent on baseline ISR-GCN2 activation, not MYC activity. We suspect that the high level of ISR-GCN2 activity in Vκ*MYC mice with early disease reflects an adaptive mechanism employed by myeloma cells to cope with cellular stress associated with transformation, but whether this is directly

mediated by MYC has yet to be determined. Nonetheless, given the demonstrated importance of MYC dysregulation in myeloma biology and progression[18–22], future studies should seek to better understand the interplay between MYC and GCN2 in this disease.

In conclusion, we present a longitudinal single-cell depiction of tumour progression using the Vκ*MYC model of myeloma. Our analysis highlights the utility of single-cell technologies for profiling the genetic and transcriptional heterogeneity of malignant cells and underscores the need for their incorporation in studies aimed at understanding cancer biology. These findings have important implications for designing targeted therapies against multiple and diverse nodes driving malignant cell subpopulations that are present in active disease. Although the hypothesis-generating portion of our study was performed using a mouse model of myeloma, the extension of our findings to myeloma patients further supports that the Vκ*MYC model is highly recapitulative of myeloma pathogenesis. Moreover, this work provides the rationale for future studies to evaluate nominated targets like GCN2 as therapeutic vulnerabilities and molecular biomarkers associated with the progression of precursor disease to overt myeloma.

## Methods

**Mice handling and bone marrow specimen processing.** Animals used in this study were housed in pathogen-free facilities at either the Montreal University Health Centre or University Health Network under the following conditions: 21 °C ambient temperature, 40–60% humidity, 12 hour dark/light cycle. All related experiments were approved by institutional Animal Care Committees and performed in accordance with the Canadian Council on Animal Care Guidelines (University Health Network Protocol #958.23, Montreal University Health Centre Protocol #2012-7242). For scRNA-seq experiments, Vκ*MYC transgenic mice[17] were cross-bred onto C57BL/KaLwRij given the latter model's increased propensity for developing spontaneous monoclonal gammopathies and bone disease[24,25]. Myeloma disease burden was monitored throughout the lifespan of the animals using serum protein electrophoresis (SPEP) of M-protein in the serum. Mice were assigned to disease groups for scRNA-seq profiling according to age and M-protein levels (not underlying biological characteristics of the tumour) as follows (Fig. 1a and Supplementary Data 1): early disease (early-MM: 27–33 weeks, no detected/trivial M-protein), intermediate-MM (int-MM, 49 weeks, trivial M-protein), and active-MM (61–74 weeks, major M-protein). To control for the possible effects of aging, non-transgenic age-matched C57BL/KaLwRij mice (55–72 weeks) were also included. At the indicated time points, mice were sacrificed and their hind leg bones dissected. BM material was extracted by flushing femurs and tibias with ice-cold PBS. Cells were then dissociated by passing through a 23-gauge needle. The resulting single-cell suspensions were subject to debris removal (35 μm cell strainer) and red blood cell lysis (ACK buffer) and then washed and resuspended in the appropriate buffer for downstream analyses.

**Disease monitoring by serum protein electrophoresis.** To monitor disease in mice, 50 μL of tail vein blood was collected into microcuvettes (Sarstedt Inc, Newton, NC, USA) and centrifuged for 5 min at 10,000 × g to separate serum. Collected serum was transferred into eppendorf tubes and stored at −20 °C until testing. SPEP was performed with 0.5–1 μL of serum using the QuickGel System (Helena Laboratories, Beaumont, Texas, USA) according to the manufacturer's instructions. Densitometric quantification of bands was performed using Image J software (NIH, Open Source) as M-protein/total protein.

**Single-cell RNA-sequencing**
*Preparation of single-cell suspensions and sequencing.* Single-cell suspensions were obtained from BM as described above and examined for cell number and viability using trypan blue and a Countess II automated counter (Thermo Fisher Scientific, Burlington, ON, Canada). Cell viability was greater than 70% for all samples described in this study. Single-cell libraries were constructed using the V2 chemistry kit from 10X Genomics (Pleasanton, CA, USA) according to the manufacturer's instructions. Libraries were sequenced on an Illumina HiSeq 2500 targeting 60,000 reads/cell. The 10X Genomics CellRanger software suite (v2) was used for processing raw sequencing reads, alignment, and to generate a digital gene expression (DGE) matrix of gene-by-cell counts. To account for the human MYC transgene in Vκ*MYC mice, the 10X Genomics GRCm38 genome reference package was supplemented with the GRCh38 MYC sequence and gene annotation. The resulting raw DGE matrices were used as input for downstream analyses using R v3.6.1.

*DGE matrix pre-processing and filtering.* Total read counts for each cell barcode were calculated with BAMTagHistogram (Drop-seq Cookbook v1-2.12[61]) and used as input for cell barcode calling using findInflectionPoint (dropbead v0.3[62]). DGE matrices containing cell-associated barcodes only were then merged for multi-sample analyses. Low-quality cells (<500 genes, <1000 transcripts (unique molecular identifier (UMI)), and/or >15% mitochondrial UMIs) and lowly expressed genes (expressed in less than 0.1% of the average number of cells per sample) were identified and removed from the analysis. Finally, suspected doublets were removed from the dataset if identified using doubletFinder[63] (v2.0.3). A summary of scRNA-seq metrics is provided in Supplementary Data 1.

*Single-cell RNA-seq data clustering and visualization.* All subsequent steps in the clustering analysis were performed using Seurat v3.2.1[64] unless otherwise stated. Log-normalized expression values were calculated for each cell $(Ei,j)$ by dividing UMI counts for gene $i$ by the sum of the UMI counts in cell $j$, to normalize for differences in coverage, multiplying by 10,000, and finally computing $\ln(Ei,j + 1)$. The 3000 most variably expressed genes were then determined using the "vst" method in FindVariableFeatures and scaled using ScaleData with regression on the proportion of mitochondrial UMIs (mt.percent). RunPCA was used to compute the top principal components using variably expressed genes. Significant principal components were identified using KneeArrower (v0.1.0) and used as input for visualization with non-linear dimensionality reduction methods (t-SNE[65], UMAP[66]), and for graph-based clustering as implemented by Seurat. Batch effect correction was applied to B cell lineage cells using Harmony[67] with group.by.vars = sequencing batch and theta = 0.5 (see Supplementary Data 1 for sequencing batch).

For the full BM scRNA-seq data set ($n = 104,880$ cells), clustering was performed using a resolution of 2.0, which clearly separated the major hematopoietic lineages present in the BM. Annotations were applied using SingleR (v1.0.6) with ref=immgen (Immgen bulk gene expression reference from highly-purified hematopoietic and immune cell types: GSE15907, GSE37448)[27]. Cell labels for scRNA-seq data were selected from the "main.label" category of the Immgen reference. Clusters where more than 50% of cells were assigned to "main.label" B cell or pro-B cell were annotated as B_PC lineage and subset for downstream analyses.

A multi-resolution clustering approach was employed for the B cell lineage scRNA-seq data set and for malignant cells from each of the seven Vκ*MYC mice with active-MM. This provided a systematic and biologically driven method to select the optimal clustering parameters using a range of resolutions (res = 0.4–1.4 for active-MM malignant cells, res = 1–2 for B cell lineage). For each resolution tested, DE analysis was performed between clusters. Only resolutions that produced clusters with a minimum specified number of differentially expressed genes were kept (5 for active-MM malignant cells, 15 for B cell lineage). The optimal clustering resolution was then selected as having the greatest median silhouette value across all clusters.

*Cell cycle scoring.* Cell cycle scores were generated using an established analysis workflow recommended in the Seurat software documentation (https://satijalab.org/seurat/v2.4/cell_cycle_vignette.html). Briefly, cell-level scores were generated based on the expression of previously published G2/M and S-phase gene signatures using the CellCycleScoring function in Seurat. G2/M and S phases were assigned based on the highest positive score, while G1-phase was assigned if both G2/M and S-phase gene scores were less than 0.

*Single-cell gene signature scoring.* Gene signature activity in single cells was estimated using the AddModuleScore function from Seurat, which calculates the average expression level of each signature on a single-cell level, and then subtracts this by the aggregated expression of control feature sets ($n = 25$). All analyzed features are binned based on averaged expression, and the control features are randomly selected from each bin. Gene sets used for signature scoring are listed in Supplementary Data 11 and when necessary, mapped to mouse orthologues using Ensembl BioMart.

*Differential expression analysis.* Differential expression (DE) analysis was performed on scRNA-seq data using the Wilcoxen rank-sum test implemented in FindAllMarkers/FindMarkers functions in Seurat. Genes were only tested if detected in over 30% of either of the two populations being compared and if their average expression exceeded 0.5-fold difference (log-scale) between the two groups, unless otherwise specified. Significance (p-value) adjustment was performed using Bonferroni correction and final results included if FDR < 0.05. The pseudo-count.use argument, which is added to the averaged expression values when calculating logFC, was adjusted to 1/number of cells in the DE analysis.

*Defining a core malignant program in Vκ*MYC tumour cells.* To define a core malignant program in Vκ*MYC mouse tumours regardless of disease stage, we performed DE analyses between malignant cells from each disease stage group and normal plasma cells. Overlapping upregulated and downregulated genes were then identified between all three disease stage groups. Core positively and negatively enriched pathways were identified from overlapping up and downregulated genes using C2, C6, and H gene sets from MSigDB (FDR < 0.05).

*Temporal patterns of gene expression throughout the progression.* To define distinct gene expression programs for each disease stage group, we performed DE analysis between the malignant cells from each sample of a given disease stage group and malignant cells from the other disease stage groups (e.g., malignant cells from EMM1 vs. malignant cells from IMM and AMM mice, see Supplementary Fig. 2b). Upregulated DE genes were removed if present in over 30% of cells from the other disease stage groups. We then identified DE genes that overlapped in all mice from a given disease group and compared the mean sample-wide expression of each gene across disease groups. Statistical comparison of means was performed using an unpaired t-test with p-values adjusted using Bonferroni correction ($P < 0.05$).

*Malignant programs of active-MM clusters.* DE analysis was performed for each active-MM mouse to define cluster-specific marker genes (as determined by resolution-optimized unsupervised clustering described above). Genes were only tested if detected in over 25% of either of the two populations being compared and if their average expression exceeded 0.25-fold difference (log-scale) compared to all other cells. Cluster-specific pathways were then identified for each set of cluster-specific markers using gene set enrichment analysis (Reactome[68], FDR < 0.05). Overlap (similarity) between enriched pathways for each cluster was determined by computing Jaccard similarity indices. Similarity groups were determined by hierarchical clustering implemented by the ComplexHeatmap R package[69]. Non-negative matrix factorization (NMF) was performed to validate the subclonal malignant programs identified above. NMF programs were identified by running the optimizeALS function from Liger[70] on the scaled dataset using $k = 20$. The top 50 features from NMF programs 1–20 were then used as input for Reactome enrichment and the result explored for similarity to programs identified by the Jaccard similarity analysis.

*Single-cell CNV analysis.* Single-cell CNV profiles were inferred using the inferCNV R package (v1.2.1)[34], which computes gene expression intensities across genomic positions from malignant cells in comparison to a set of reference cells (normal plasma cells). Input data including raw counts matrix, annotations file, and positions file (provided in SourceData_Fig. 3.xlsx) were prepared as recommended by inferCNV authors. The algorithm was run with the following arguments: cutoff=0.1, HMM_type = 'i3', cluster_by_groups=TRUE, denoise=TRUE, HMM = TRUE. CNV-level subpopulations were determined using analysis_mode = 'subclusters', which attempts to partition cells into groups having consistent patterns of CNVs. Enrichment analysis of del(5) cells was performed using marker gene analysis (logfc = 1.0, FDR < 0.05) and gene set enrichment (Reactome[68], FDR < 0.05) using CNV-partitioned cells with predicted chr5 state of 1 compared to all other malignant cells. Mouse-human orthologues were determined by mapping mouse chr5 genes (GRCm38.p6) to the full human genome (GRCh38.p13) using Ensembl BioMart (Ensembl Genes 101).

**Acquisition and analysis of public genomic datasets**. All publicly available patient data were downloaded from the Gene Expression Omnibus. scRNA-seq count data from Ledergor et al.[12] (GSE117156) were filtered according to parameters specified in the original manuscript and processed using Seurat. Signature scores were calculated using the Seurat AddModuleScore function for each gene set. Raw data from Chng et al.[43] (GSE6477) were log2-quantile normalized prior to scoring. RNA-seq data for HMCLs was obtained from the Keats Lab repository using https://www.keatslab.org/data-repository (HMCL66_Gene_Expression_FPKM). Signature scores for Chng et al. and HMCL expression data were calculated using the mean scaled expression of gene set genes.

**Human myeloma cell lines**. Experiments using human myeloma cell lines (HMCLs) XG7, XG6, MM1S, MY5, JJN3, OPM2, U266, AMO1, and RPMI-8226 were performed in L-glutamine free Iscove's Modified Dulbecco's Medium (IMDM, Gibco, Grand Island, NY, USA) supplemented with 5% fetal bovine serum (FBS) (Hyclone, Logan, UT, USA), 100 μg/ml penicillin and 100 μg/ml streptomycin (Gibco, Grand Island, NY, USA). Cell lines were maintained routinely in a humidified chamber at 37 °C and 5% carbon dioxide. Transplantable Vκ12598 cells[71] were generously provided by Dr. Marta Chesi.

**GCN2 guide RNA cloning and lentivirus production**. To generate guide RNA (gRNA) targeting GCN2 expression, we employed the LentiCRISPR v.2 vector system (Addgene, Watertown, MA, USA). Constructs were designed to maximize on-site specificity and minimize off-target activity using a publicly-available online CRISPR design algorithms[72]. RNA guides were cloned into LentiCRISPR v.2 puromycin-resistant backbone and selected by ampicillin resistance in *Stbl3* bacteria according to the Lentiviral CRISPR Toolbox standard protocol from the Zhang Lab[73,74]. GCN2 guide and control RNA lentiviral particles were produced by transfecting HEK293T cells (obtained from ATCC) using the Lipofectamine 3000 transfection reagent (Thermo Fisher Scientific, Burlington, ON, Canada). The medium was replaced with IMDM medium containing 10% FBS 12 h after transfection, and medium containing viral particles was collected 48 and 72 h after initial transfection. Media was filtered using a 0.45 μM filter, aliquoted, and stored at −80 °C. DNA oligonucleotides used for GCN2 knockout experiments are described in Supplementary Table 1.

**Generation of GCN2$^{-/-}$ knockout HMCLs**. RPMI-8226 and OPM2 cells were stably transduced with lentiviral LentiCRISPR v.2 expressing VAS9 gene and gRNAs targeting GCN2 or control lentivirus for 24 h at 37 °C in antibiotic-free IMDM with polybrene. Lentivirus media was removed and replaced with fresh IMDM media including antibiotic (1% penicillin-streptomycin) and incubated for 48 h. Stably transduced cells were then selected using 2 μM puromycin for 3 days and GCN2 knockouts validated by western blotting using anti-GCN2 (Cell Signalling Technologies, cat.3302, 1:1000 dilution), anti-TOM40 (Proteintech, cat.18409, 1:1000 dilution) and anti-β-Tubulin (Cell Signalling Technologies, cat.2146, 1:1000 dilution). HRP-conjugated anti-rabbit was used as a secondary antibody (Cytiva, cat.NA934, 1:1000 dilution). GCN2 knockout and control vector HMCLs were cultured in IMDM media without L-glutamine and after 5 days, viability was analyzed using trypan blue (0.4% Trypan Blue Solution, Gibco, Grand Island, NY, USA). Two independent constructs were used for CRISPR knockout experiments and viability measurements were obtained in duplicate.

**In vitro analysis of GCN2iB in HMCLs and Vκ12598**. The GCN2 inhibitor GCN2iB[44] was obtained from MedChemExpress (Monmouth Junction, NJ, USA) and dissolved in DMSO as per the manufacturer's recommendations. At the time of experiments, stock solutions were thawed and diluted in culture medium (DMSO vehicle controls were prepared at the same volumes, such that the final solutions contained the same percentage DMSO). Cell death of HMCLs was assessed by flow cytometric analysis of apoptosis using the FITC Annexin V Apoptosis Detection Kit (BD Biosciences, cat.556547) according to the manufacturer's instructions. Cell death of Vκ12598 cells was assessed by flow cytometric analysis of CD138+/B220- cells using CD138-APC (BD Biosciences, cat.558626, clone.281-2) and B220-PE (BD Biosciences, cat.553089, clone.RA3-6B2) according to the manufacturer's instructions. All samples were analyzed on a BD FACS Canto II flow cytometer (BD Biosciences, San Diego, CA, USA), with data collected by FACSDiva v8.0.1 and results analyzed using Flowjo v10.7.1. Cell viability after treatment with GCN2iB was assessed by MTT assay (Roche Molecular Biochemicals, Boehringer, Germany) according to the manufacturer's instructions. Dye absorbance was read at 570 nm (reference wavelength; 650 nm) using the OptiMax microplate reader (Molecular Devices, Sunnyvale, CA, USA).

**Statistics**. Statistical tests performed are indicated in figure legends. The sample size was determined by the availability of subjects for scRNA-seq, but a minimum of three samples for each disease stage group was decided upon upfront to capture biological heterogeneity between tumours. Measurements were taken from distinct samples unless otherwise stated.

**Reporting summary**. Further information on research design is available in the Nature Research Reporting Summary linked to this article.

## Data availability

The scRNA-seq data generated in this study have been deposited as raw bam files and as processed gene expression matrices at the National Centre for Biotechnology Information Gene Expression Omnibus with accession numbers SRP214856 and GSE134370. Source data for all figures are also provided for this paper. Previously published data sets are available without restriction from the Gene Expression Omnibus (GSE117156 and GSE6477). Source data are provided with this paper.

## Code availability

Code supporting this study is available at https://github.com/pughlab/scVkMYC_mPC.

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

## Acknowledgements
This work was supported by the Government of Canada through a Canadian Institute for Cancer Research Project Grant (CIHR #159465). Additional funding was provided by the Princess Margaret Cancer Foundation. Infrastructure support was received from the Canada Foundation for Innovation—John R. Evans Leaders Fund [CFI #38401] and the Ministry of Colleges and Universities Ontario Research Fund—Research Infrastructure Program. D.C.C. was supported by an Ontario Graduate Scholarship, the David Rae Graduate Student Scholarship from the University of Toronto, and a Graduate Fellowship in Cancer Research from the Princess Margaret Hospital Foundation. M.C. was supported by the National Cancer Institute grant CA234181. T.J.P. holds the Canada Research Chair in Translational Genomics and is supported by a Senior Investigator Award from the Ontario Institute for Cancer Research and the Gattuso-Slaight Personalized Cancer Medicine Fund. We gratefully acknowledge the animal facility staff at the Princess Margaret Animal Resource Centre and at the Montreal University Health Centre for their technical assistance with animal protocols. We thank the staff of the Princess Margaret Genomics Centre (www.pmgenomics.ca, Troy Ketala, Neil Winegarden, Julissa Tsao, Nick Khuu) and Bioinformatics and High-Performance Computing Core (Carl Virtanen, Zhibin Lu, and Natalie Stickle) for sample coordination and their expertize in generating the single-cell sequencing data used in this study.

## Author contributions
D.C.C., T.J.P. and S.T. conceived the idea and designed the study. D.C.C. prepared bone marrow samples for scRNA-sequencing and performed all related data analysis. D.C.C. obtained and analyzed all publicly available single-cell and bulk RNA-seq data sets. D.C.C., S.P.T., Z.L., and E.N.W. optimized and conducted in vitro experiments. M.S., D.W., and X.F.H. provided support for all transgenic mouse experiments. T.J.P., S.T., D.W., M.C., P.L.B., and M.S. provided guidance in experimental protocol design. L.M.R. assisted with data analysis. L.M.R., M.C., P.L.B., M.S., T.J.P., and S.T. provided guidance in data analysis and interpretation of the results. D.C.C. drafted the manuscript, with significant input from T.J.P. and S.T. All authors reviewed the manuscript prior to submission.

## Competing interests
The authors declare no competing interests.
