## [Peer review file · Nature Communications]

REVIEWER COMMENTS

Reviewer #1 (Remarks to the Author): Expert in myeloma genomics

Croucher et al. present an elegant and meaningful paper where they apply single-cell RNA sequencing on the Vk*MYC mouse model at different time windows.

The Vk*MYC mouse model is quite unique in cancer. In fact, this model allows the investigation of both the spontaneous clonal/subclonal selection and the immune-microenvironment changes involved in multiple myeloma pathogenesis. The Authors provide a clear story showing how genomic and transcriptomic drivers can be detected very early on time, and how the tumor progressed through the competition of multiple subclones. Finally, investigating the intra-tumor heterogeneity at the single-cell level, they identify new potentially druggable pathways and they validate these observations on both published data and cell lines.

The methods and study design are robust, and the data are convincing.

Overall, I have only minor comments and suggestions:

- Methodologically and biologically speaking, the distinction between early and intermediate disease is not fully clear. Authors divide these two groups using "age" and "monoclonal protein size". However, from the presented biological investigations, it seems that these two groups are quite similar (e.g. Figure 1b, 1h, 1I; Figure 2a; Suppl. Figure 1). For example, looking at Figure 1b, it seems that two early-MM samples have an M-protein similar to the control, while three have an M-protein similar to the intermediate-MM group. In Figure 1h, the normal plasmacell infiltration is identical in all but one case with early- and intermediate-MM(EMM2). Can this heterogeneity/similarity suggest that the Vk*MYC mouse progression has an exponential shape (i.e., small changes over time, until the final acceleration at full progression)? Similar to what we see in myeloma patients where a "myeloma-to-be clone" proceed the symptomatic progression of several years? Would it be possible that some myeloma progressed faster than others? Or have less M-protein? If so, how this can be taken into account, and how the similarity between early-MM and intermediate-MM affect some of the analysis?

- Following up on the previous point, it is not completely clear why we have intermediate gene expression peaks (Figure 2d and 2e), and which is the biological rationale of these changes. For example, which is the advantage to have a peak of distinct pro angiogenetic genes in intermediate Vk*MYC MM, but not in active MM? While reduced or increased expression over time of certain transcripts makes sense, the peaks reported are a bit more difficult to understand from the cancer perspective.

Being so similar, it is also unclear which are the key differences between the three groups: early, intermediate, and myeloma. This re-connects with what is mentioned above. It is hard to see differences between these three groups. it might be that the clone is fairly conserved and just need time to progress and to subvert the immune surveillance.

- The copy number analysis is one of the most interesting parts of this paper. While the heterogeneity is evident by all the informative Figures, I wonder how many cells are needed to create a distinct subclone. Some "clusters/subclones" in Figure S3 look quite small, and it might be difficult to estimate their expression and CNV uniqueness.

- The CNV landscape between different groups seems to be similar. This would suggest that the key aneuploidies in the Vk*MYC mouse model are acquired very early on time, similarly to what has been recently described in humans by Oben et al. Nat Comm 2021 and Rustad et al. Nat Comm 2020. I would definitely stress this part being extremely relevant from the translational perspective. I completely agree that these data recapitulate what is seen in humans, and I would suggest further expand what is discussed on page 15, lines 410-414. Specifically, Bolli et al. reported the "static

model" as a way through which a fraction of SMM progressed without any significant changes in their clonal and subclonal composition. However, here, what is striking is that, despite the static progression, at the single-cell level, there is a high competition between subclones, otherwise undetectable by bulk sequencing. This observation clearly supports the idea that, despite the "bulk" stability, positive selection is constantly active in multiple myeloma, as recently suggested by Diamond et al. Leukemia 2021.

- Methodologically speaking it is not completely clear how they inferred CNV and expression data. I agree that the microenvironment and epigenome can affect the expression profile, but the observed heterogeneity might be also driven by other features, such as different cell cycle stages or uncertainty in the clustering. In Figure 4 Authors reported different cell cycle cell stages. I wonder how this affects the integration of CNV and RNA expression data. For example, in Figure S3, AMM1 seems to have 4 clear CNV and 13 RNA clusters. From Figure 3, it seems that AMM1 red and blue clusters overlap, while green and orange tend to define two distinct clusters. Would it be possible that small CNVs might not be enough to significantly affect gene expression? Or could the different cell cycle phases be responsible for the higher number of clusters in RNA data compared to CNV? In AMM2 it seems that you have two dominant clusters (one with 5q loss and one without), plus two small clusters. According to the RNA data, it seems that the two main clusters are quite well divided. Considering single-cell uncertainty, major copy number changes, and eventually cell cycle, It might be that the concordance between CNV and RNA is not so bad as reported in the manuscript.

- Regarding the chr5 deletion, the word "universal" sounds a bit excessive considering the sample size. Also, looking at the CNV clustering, it seems that chr5q loss is acquired as a secondary hit. In fact, only in AMM5 all cells have chr5 loss. In most of the cases, the same lesion is lost only in a fraction, suggesting that something else preceded chr5 loss acquisition and that Vk*MYC clones can expand and be selected also without chr5 loss. The observation is still extremely informative and clearly shows the advantage of single-cell technologies in this specific setting.

- I think that, if the authors wanted to link 5q loss and other mouse lesions to humans they should do it in an extensive and detailed way (e.g., listing genes). I personally don't feel that this link is needed in this paper, and I would consider removing that part (lines 234-240).

- Is the GCN2 pathway activity correlated with the size of the M-protein?

- How did the Author define "focal gain" (page 8, line 213)?

Francesco Maura, MD
University of Miami

Reviewer #2 (Remarks to the Author): Expert in single-cell RNA-seq and cancer genomics

In "Longitudinal single-cell analysis of a myeloma mouse model identifies early emergence of subclonal molecular programs associated with progression", the authors investigated tumour progression of myeloma using longitudinal single-cell transcriptomics data from a genetic mouse model. The analyses showed extensive genetic and transcriptional heterogeneity of the malignant cells. The main findings are a shared subpopulation across samples with a ISR-GCN2 program related to protein translation and amino acid deprivation and an early subclonal loss of chr5 present in all tumours.

The study was well designed, however the analyses are a bit superficial in some parts. The authors show extensive transcriptomic heterogeneity, but what the specific subpopulations/states are and what drives this heterogeneity and how this heterogeneity drives tumour progression is not investigated in detail.

Specific comments:

The authors show that cell state heterogeneity arises largely independently of CNV variation which has also been shown previously in Ledergor et al. (Nature Medicine, 2018) and are suggesting that there are additional non-genetic mechanisms driving tumour progression, for example epigenetic programs or the microenvironment. This is very interesting, but the authors did not go further to investigate the mechanisms that drive this heterogeneity.

Related to this transcriptomic heterogeneity, they identify 3-13 clusters across patients, however, they do not go into more detailed analyses to investigate the transcriptomic heterogeneity. What are those subpopulations? Are there shared clusters across samples? Are there clusters enriched in early or late samples, are there specific cellular states promoting tumour progression? Does the diversity increase or decrease over progression?

If the progression is driven by non-genetic mechanisms, this could indicate plasticity. Are there continuous changes across tumour progression, are cells transitioning from one state to another during tumour progression? If yes, are there certain transcription factors driving this transition?

The authors found only two shared programs, cell cycle and ISR-GCN2 program. Since the results do not show many DE genes between each stage of progression, another approach to look for shared molecular programs across samples would be using non-negative matrix factorisation and then looking how those programs/modules change along progression.

The validation using GCN2 inhibitor is convincing, but it would be good to see survival analyses using big cohorts like TCGA to support the prognostic relevance of the ISR-GCN2 pathway.

Reviewer #3 (Remarks to the Author): Expert in myeloma mouse models and therapy

In the manuscript by Croucher et al., the authors report a disease spectrum-spanning, single-cell transcriptional analysis of the V κ *MYC mouse model of myeloma, to study malignant cell evolution in the context of a uniform transforming path. Moreover, the authors show that the ISR-GCN2 pathway activity is progressively activated during progression in myeloma patients and is highly active in a subset of newly diagnosed MM samples. Indeed, chemical and genetic perturbation of GCN2 inhibits myeloma cell survival in vitro, which correlated with baseline ISR-GCN2 activity.

The experiments are thoughtfully designed and the conclusions are supported by the results presented. However, I have some comments:

- 1) In Figure 5, the authors show that the ISR-GCN2 pathway supports myeloma cell survival in vitro experiments, according to baseline ISR-GCN2 activity.
 - In panel A, the authors show a progressive increase of ISR-GCN2 expression during progression in myeloma patients. The authors should evaluate if there is a direct functional interaction between MYC and GCN2. The authors should explore if MYC directly promotes GCN2 transcription.
 - The activity of GCN2-inhibitor does not seem related to MYC expression in tested MM cell lines. Indeed, the authors show no activity of GCN2-inhibitor in MYC overexpressing MM1S cells, while they demonstrate apoptosis induction in MYC-null U266 cells after GCN2-inhibitor. Furthermore, very modest activity is reported on V κ 12598 tumor cells. How the authors could explain these data?
 - The authors should check the effect of GCN2 knockdown in GCN2-i insensitive MM1S cell survival.
- 2) The authors report the activation of processes related to chromosomal instability in ^{SEP} cells with chr5 deletion as compared to wild-type chr5 (chr5WT) cells. Could the authors evaluate the differential occurrence of specific mutational signatures (PMID: 23945592) in chr5del vs chr5WT, to understand

the potential role of error-prone DNA repair pathways, such as Alternative Non-Homologous End Joining Repair, in chr5del malignant plasma cells?

Response to Reviewers – Croucher et al.

We thank the reviewers for their thoughtful and detailed review of our manuscript. We have responded in-kind to address their comments below. Reviewer comments are in *red, italicized text*, followed by our response in black text and any resulting manuscript changes (verbatim) in *blue, underlined text* (blue, non-underlined text may be provided to show context of the change within the revised manuscript). Additional figures supporting our responses can be found at the end of this document.

Reviewer #1 (Expert in myeloma genomics): Remarks to the Author

Croucher et al. present an elegant and meaningful paper where they apply single-cell RNA sequencing on the $V\kappa^$ MYC mouse model at different time windows.*

The $V\kappa^$ MYC mouse model is quite unique in cancer. In fact, this model allows the investigation of both the spontaneous clonal/subclonal selection and the immune-microenvironment changes involved in multiple myeloma pathogenesis. The Authors provide a clear story showing how genomic and transcriptomic drivers can be detected very early on time, and how the tumor progressed through the competition of multiple subclones. Finally, investigating the intra-tumor heterogeneity at the single-cell level, they identify new potentially druggable pathways and they validate these observations on both published data and cell lines.*

The methods and study design are robust, and the data are convincing. Overall, I have only minor comments and suggestions.

We thank the reviewer for their strong positive assessment of our manuscript and for acknowledging the quality and novelty of our work. We have addressed the reviewer's minor comments and suggestions below.

Reviewer 1, Comment 1)

Methodologically and biologically speaking, the distinction between early and intermediate disease is not fully clear. Authors divide these two groups using “age” and “monoclonal protein size”. However, from the presented biological investigations, it seems that these two groups are quite similar (e.g. Figure 1b, 1h, 1i; Figure 2a; Suppl. Figure 1). For example, looking at Figure 1b, it seems that two early-MM samples have an M-protein similar to the control, while three have an M-protein similar to the intermediate-MM group. In Figure 1h, the normal plasma cell infiltration is identical in all but one case with early- and intermediate-MM (EMM2). Can this heterogeneity/ similarity suggest that the $V\kappa^$ MYC mouse progression has an exponential shape (i.e., small changes over time, until the final acceleration at full progression)? Similar to what we see in myeloma patients where a “myeloma-to-be clone” proceed the symptomatic progression of several years?*

We agree with the reviewer's observation that there is only a subtle difference between early-MM and intermediate-MM $V\kappa^*$ MYC mice when defined by disease burden (**Fig. 1b, Fig. 1i**). However, our data support that there are clear differences in gene expression between the early-MM and intermediate-MM disease stage groups, despite having similar disease burden levels as measured by M-protein. These findings are presented in **Fig. 2c-e** and **Fig. 4c** of the manuscript as well as described in the text as follows:

[Lines 176-180] By employing DE analysis, we defined the temporal expression patterns that are specific to malignant cells from each stage of progression (see Methods, Supplementary Fig. 2b-

c, Supplementary Table 5). This revealed 21 genes with expression levels that changed significantly throughout progression (Fig. 2c-e) and whose longitudinal pattern of expression coincided with one of three different groups.

[Lines 314-317] Not only was the ISR-GCN2 pathway upregulated in malignant cells from all disease groups compared to normal plasma cells, the highest level of ISR-GCN2 activation was observed in malignant cells from mice with the earliest stages of disease (early-MM) (Fig. 4c).

The methodology of how we defined our disease stage groups is based on standard practices in myeloma patients where diagnosis of precursor disease stages (MGUS and SMM) is based on clinical measures of disease burden, not on underlying biological characteristics of the tumour. We have now made this explicit point in the Methods:

[Line 542-544] Mice were assigned to disease groups for scRNA-seq profiling according to age and M-protein levels (not underlying biological characteristics of the tumour)

We do agree that progression, if measured by disease burden, does appear to have an exponential shape, which is another similarity between progression in the V_{κ} *MYC model and in human patients. We have included this observation in the revised manuscript:

[Lines 104-107] Disease stage groups were defined based on serum M-protein levels (Fig. 1b) and age (Fig. 1c). While the distinction between early and intermediate disease based on these criteria was subtle, mice with active disease demonstrated a significantly higher serum M-protein compared to all other mice (Fig. 1b), supporting an exponential pattern of progression.

Would it be possible that some myeloma progressed faster than others? Or have less M-protein? If so, how this can be taken into account, and how the similarity between early-MM and intermediate-MM affect some of the analysis?

It was our intention to employ a model driven consistently by the same transforming event such that progression should, in theory, proceed consistently across all mice. Thus, our disease stage groups would represent a sampling of the biological characteristics at their respective time point. However, our data supports that stochastic events can lead to molecular divergence, despite sharing a transforming event, and it is possible that this may in turn lead to differences in the trajectory of progression for each mouse, as suggested by the reviewer. Exploring the molecular changes that occur throughout progression within an individual mouse via serial sampling may provide insight into this possibility, which could be addressed in future work upon optimization of serial sampling protocols in the mice. We have introduced this potential future direction in the Discussion as follows:

[Lines 409-415] To better understand if and how these features contribute to differences in disease trajectories and rates of progression, future studies could consider exploring molecular changes that influence disease progression within an individual mouse via serial sampling. This approach could also enable the characterization of continuous changes such as cancer cell plasticity that may contribute to intratumoural heterogeneity and could further remove biological variability between mice that may have impacted our differential expression analysis across disease stages.

[Lines 432-437] It is possible that minor differences in disease burden between early-MM and int-MM and/or biological variability between mice may account for the lack of broad differences in gene expression between these disease stage groups. However, our observations are consistent with those reported by Storti *et al.* who performed bulk microarray profiling of CD138-selected

cells from patient BM and similarly found that the transcriptomic profile of myeloma cells remains substantially unchanged throughout progression.¹

Reviewer 1, Comment 2)

*Following up on the previous point, it is not completely clear why we have intermediate gene expression peaks (Figure 2d and 2e), and which is the biological rationale of these changes. For example, which is the advantage to have a peak of distinct pro angiogenic genes in intermediate Vk*MYC MM, but not in active MM? While reduced or increased expression over time of certain transcripts makes sense, the peaks reported are a bit more difficult to understand from the cancer perspective. Being so similar, it is also unclear which are the key differences between the three groups: early, intermediate, and myeloma. This re-connects with what is mentioned above. It is hard to see differences between these three groups. It might be that the clone is fairly conserved and just need time to progress and to subvert the immune surveillance.*

To explain the finding of this “Peak-Intermediate” gene expression pattern, we refer the reviewer to the pattern of ISR-GCN2 activity during disease progression, which also showed a similar pattern (peak in early-MM, reduced in int-MM, see **Fig 4c. panel 1**). Our interpretation of this finding has been elaborated upon in the manuscript as follows:

[Lines 322-325] Given the role of MYC in promoting protein synthesis, these results put forth a model of disease in Vk*MYC mice whereby ISR-GCN2 activation occurs early in disease pathogenesis to tolerate cellular stress caused by MYC activation and the resulting excessive protein translation (Supplementary Fig. 5c).

The observation that ISR-GCN2 pathway activation is subclonal in active-MM mice, but has the highest activity in early-MM mice suggests that molecular variability in advanced disease may reflect residual programs from precursor disease. To further support this hypothesis, we performed additional analyses using our scRNA-seq data to evaluate whether the dominant malignant program in intermediate-MM mice (i.e. **Fig 2e**, “Peak at intermediate-MM” genes) could be detected in any of the transcriptional clusters from active-MM mice. Indeed, these “Peak-Intermediate” genes were highly expressed in only a subset of cells from active-MM mice (**Revision Fig. 1**). Moreover, the pathways associated with these clusters (AMM1_C9, AMM3_C4, AMM4_C8) were enriched with terms related to immune processes (Program C from **Fig. 4a** and **Revision Fig. 2**), consistent with our observation that many of the “Peak-Intermediate” genes in **Fig. 2e** play a role in immune processes. To make our interpretation of these findings more clear, we have updated the manuscript as follows:

- 1) We created a schematic that illustrates the model of progression supported by our data. This schematic is now included as **Fig. 5** in the main manuscript.
- 2) We have created two new Supplementary Figures, **Supplementary Fig. 5h** and **Supplementary Fig. 4**, (**Revision Fig. 1** and **Revision Fig. 2**, respectively).
- 3) We have revised the manuscript Abstract [**Lines 23-25**], Introduction [**Lines 86-91**], Results [**Lines 328-340**] and Discussion [**Lines 452-458**] to highlight these findings.

Abstract: [Lines 23-25] We detect intratumoural heterogeneity driven by transcriptional variability during active disease and show that subclonal expression programs are enriched at different times throughout early disease.

Introduction: [Lines 86-90] These analyses reveal that the malignant cell compartment of mice with early disease is comprised of multiple CNV-driven subpopulations and is enriched for transcriptional programs identified as subclonal in mice with active-MM. Moreover, pathway-level analyses revealed that one of these subclonal pathways relates to an adaptive program in malignant cells involving GCN2 and the integrated stress response pathway.

Results: [Lines 331-340] The observation that ISR-GCN2 pathway activation is subclonal in active-MM mice, but has the highest activity in early-MM mice suggests that molecular variability in advanced disease may reflect residual programs from precursor disease. To support this model of progression, we scored the transcriptional clusters in active-MM mice using highly expressed genes in int-MM mice from our longitudinal analysis (“Peak-Intermediate”, Fig. 2e). In doing so, we found that “Peak-Intermediate” genes were highly expressed in only a subset of cells from active-MM mice (Supplementary Fig. 5h) and that the pathways associated with these clusters were also related to immune processes (Supplementary Fig. 4, Program C). Thus, these data suggest that gene expression programs that once defined the landscape of malignant cells in precursor disease stages become subclonal in advanced disease as tumours diversify (Fig. 5).

Discussion: [Lines 459-465] The persistence of certain subclonal CNVs, such as loss of chr5, throughout progression supports the notion that evolution of disease in the V κ *MYC model is shaped by positive selection, as recently suggested by Diamond *et al.*² However, our findings add an additional layer to these models whereby transcriptional divergence is a characteristic feature of progression that reflects previously dominant molecular programs that may have been selected against during progression (Fig. 5). This supports the use of preventative treatment strategies in the precursor setting, when tumours are less molecularly complex and potentially easier to target.

Reviewer 1, Comment 3)

The copy number analysis is one of the most interesting parts of this paper. While the heterogeneity is evident by all the informative Figures, I wonder how many cells are needed to create a distinct subclone. Some “clusters/subclones” in Figure S3 look quite small, and it might be difficult to estimate their expression and CNV uniqueness.

We computed the cell counts for each cluster and compared the results for CNV- vs. gene expression-driven subpopulations:

Sample	CNV-driven (# of clusters)	GE-driven (# of clusters)
AMM1	10-979 (5)	10-427 (13)
AMM2	11-352 (7)	28-393 (4)
AMM3	1-704 (6)	60-474 (7)
AMM4	4-386 (8)	34-189 (9)
AMM5	1-83 (8)	22-153 (4)
AMM6	7-175 (7)	44-97 (3)
AMM7	11-128 (7)	31-75 (6)

We have now added a summary of these CNV- and gene expression-driven clustering data in the legend of **Fig. 3a** and **Fig. 3b**. These data show that the minimum number of cells forming a unique cluster is similar between CNV-driven vs. gene expression-driven subpopulations, with slightly lower minimum cell numbers in CNV-driven clustering. This is likely explained by the fact that the clustering algorithms

implemented in the two analyses are different, which we have now acknowledged in the manuscript discussion as a potential caveat.

[Lines 421-424] We acknowledge the possibility that these differences may be a result of different clustering algorithms used to group the data. However, it is notable that our finding of lack of overlap between some CNV-level subpopulations and transcriptionally-defined subclones is consistent with the recent findings from Fan *et al.*³

Reviewer 1, Comment 4)

*The CNV landscape between different groups seems to be similar. This would suggest that the key aneuploidies in the Vκ*MYC mouse model are acquired very early on time, similarly to what has been recently described in humans by Oben *et al.* Nat Comm 2021 and Rustad *et al.* Nat Comm 2020. I would definitely stress this part being extremely relevant from the translational perspective. I completely agree that these data recapitulate what is seen in humans, and I would suggest further expand what is discussed on page 15, lines 410-414. Specifically, Bolli *et al.* reported the "static model" as a way through which a fraction of SMM progressed without any significant changes in their clonal and subclonal composition. However, here, what is striking is that, despite the static progression, at the single-cell level, there is a high competition between subclones, otherwise undetectable by bulk sequencing. This observation clearly supports the idea that, despite the "bulk" stability, positive selection is constantly active in multiple myeloma, as recently suggested by Diamond *et al.* Leukemia 2021.*

We thank the reviewer for their support of this particular finding. We have taken their suggestion to further expand upon the discussion as to how our data recapitulate what is seen in human myeloma patients, including the addition of their aforementioned citations in the revised manuscript. As discussed in Comment 2 above, we have added a schematic to the manuscript (**Fig. 5**) to depict a proposed model for tumour heterogeneity throughout disease progression.

[Lines 213-216] The presence of multiple cell subpopulations with distinct CNV profiles was evident at all disease stages. This therefore suggests that genomic diversification of tumours begins early in the evolution of myeloma tumours, similar to what has been recently described in humans^{4,5}, and continues throughout progression.

[Lines 452-465] Nonetheless, our inferred CNV analysis supports the presence of distinct malignant cell populations at all stages of disease and further shows that these subclones persist throughout disease progression. This pattern of progression is consistent with the "static model" recently proposed by Bolli *et al.*⁶ In this model, subclonal architecture is retained as disease progresses from SMM to MM, such that progression reflects the time it takes to accumulate sufficient disease burden, rather than the acquisition of an additional oncogenic hit. Thus, findings from our study may be most applicable to tumours from SMM patients that follow this evolutionary path to progression. The persistence of certain subclonal CNVs, such as loss of chr5, throughout progression supports the notion that disease evolution in the Vκ*MYC model is shaped by positive selection, as recently suggested by Diamond *et al.*² However, our findings add an additional layer to these models whereby transcriptional divergence is a characteristic feature of progression that reflects previously dominant molecular programs that may have been selected against during progression (Fig. 5). This supports the use of preventative treatment strategies in the precursor setting, when tumours are less molecularly complex and potentially easier to target.

Reviewer 1, Comment 5)

Methodologically speaking it is not completely clear how they inferred CNV and expression data.

We apologize that our description of inferred CNV analysis was unclear. We now refer readers to a detailed description of how the InferCNV algorithm was specifically used in our study to define CNV-driven subpopulations:

[Line 201-203] To evaluate whether malignant cells demonstrate heterogeneity driven by somatic genome alterations, we inferred CNVs from the gene expression profiles of our scRNA-seq data using the InferCNV⁷ algorithm (see Methods for details).

We have also added an explanation of our analysis that evaluated the intersection of CNV-driven vs. RNA-driven subpopulations:

[Lines 239-248] Therefore, we mapped CNV-defined subpopulations to transcriptional clusters (Fig. 3b) determined by unsupervised clustering of scRNA-seq data from Vκ*MYC mice with active-MM. Similar to our CNV analysis, transcriptional clustering of the data revealed that the malignant cell compartment of Vκ*MYC mice is comprised of multiple discrete molecular subpopulations per mouse (3-13 transcriptional clusters, Fig. 3b). The fact that samples with a similar number of cells did not have the same number of transcriptional clusters (e.g. AMM2 and AMM4, AMM6 and AMM7) supports that the range in the number of transcriptional clusters is not just an artifact of differences in the number of cells profiled across samples. We then evaluated whether this transcriptional heterogeneity was driven by the subclonal CNVs inferred above by exploring the distribution of CNV-driven subpopulations within each transcriptional cluster.

I agree that the microenvironment and epigenome can affect the expression profile, but the observed heterogeneity might be also driven by other features, such as different cell cycle stages or uncertainty in the clustering. In Figure 4 Authors reported different cell cycle cell stages. I wonder how this affects the integration of CNV and RNA expression data. For example, in Figure S3, AMM1 seems to have 4 clear CNV and 13 RNA clusters. From Figure 3, it seems that AMM1 red and blue clusters overlap, while green and orange tend to define two distinct clusters. Would it be possible that small CNVs might not be enough to significantly affect gene expression? Or could the different cell cycle phases be responsible for the higher number of clusters in RNA data compared to CNV? In AMM2 it seems that you have two dominant clusters (one with 5q loss and one without), plus two small clusters. According to the RNA data, it seems that the two main clusters are quite well divided. Considering single-cell uncertainty, major copy number changes, and eventually cell cycle, It might be that the concordance between CNV and RNA is not so bad as reported in the manuscript.

Our findings support that subclonal CNV events can, in some cases, affected gene expression to generate distinct transcriptional clusters upon unsupervised clustering. However, our analysis found that most subclonal CNV events were distributed across multiple transcriptional clusters. We interpret this to mean that there are other mechanisms driving transcriptional heterogeneity in the malignant cell compartment. To more clearly demonstrate these points, we have re-phrased our results sections in the revised manuscript as follows:

[Lines 246-256] We then evaluated whether this transcriptional heterogeneity was driven by the subclonal CNVs inferred above by exploring the distribution of CNV-driven subpopulations within each transcriptional cluster. In doing so, we observed instances of majority CNV-driven transcriptional clusters (Fig. 3c-d). For example, in AMM1, transcriptional cluster 4 was largely comprised of cells from CNV subpopulation 3, defined by del(5) and del(12). Similarly, in AMM4, transcriptional cluster 0 was largely comprised of cells from CNV subpopulation 1, defined by subchromosomal gain of chromosome 9. This supports that subclonal CNVs can have

a significant effect on the formation of distinct transcriptional clusters. However, the majority of CNV-driven subpopulations were distributed across several transcriptional clusters (Fig. 3d), suggesting that transcriptional variability must be driven by additional sources beyond subclonal CNV events.

We have also added a corresponding colour legend for each CNV-driven subpopulation in **Fig. 3d** and **Supplementary Fig. 3a** so that readers can more clearly map subclonal CNVs to their respective transcriptional clusters.

The reviewer poses an interesting question regarding how differences in cell cycle stages may affect the mapping of CNV and gene expression data. We explored this question by defining the CNV subclones present within proliferative clusters and compared this to non-proliferative clusters. As shown in **Revision Fig. 3**, proliferative clusters were comprised of multiple CNV subclones supporting that the gene expression changes associated with proliferation are more profound than those induced by CNV events. Despite this however, CNV-driven subpopulations still distribute across several non-proliferative transcriptional clusters, supporting that some transcriptional variability can be explained by CNV-driven expression changes, but that CNVs do not account for all transcriptional variability. We have included this observation in the revised manuscript as **Supplementary Fig. 5b** and in the main text as follows:

[Lines 295-297] Finally, these proliferative clusters were comprised of multiple CNV subclones supporting that the gene expression changes associated with proliferation were more profound than those induced by CNV events (Supplementary Fig. 5b).

Reviewer 1, Comment 6)

*Regarding the chr5 deletion, the word “universal” sounds a bit excessive considering the sample size. Also, looking at the CNV clustering, it seems that chr5q loss is acquired as a secondary hit. In fact, only in AMM5 all cells have chr5 loss. In most of the cases, the same lesion is lost only in a fraction, suggesting that something else preceded chr5 loss acquisition and that Vκ*MYC clones can expand and be selected also without chr5 loss. The observation is still extremely informative and clearly shows the advantage of single-cell technologies in this specific setting.*

We agree that the word universal may be overstated so we have changed the text as outlined below. We have also removed this finding from the abstract to de-emphasize.

[Lines 472-474] We also show that loss of chr5 is a highly recurrent event in Vκ*MYC tumours and more common than previously reported by Chesi *et al.* who found an incidence of approximately 50%^{8,9}.

We agree that chr5 loss is almost certainly secondary to the initial event of MYC dysregulation that initiates transformation in this model. A more explicit explanation of this has now been included in the manuscript discussion section as outlined below and in the newly created **Fig. 5**.

[Line 470-472]. It is therefore highly likely that loss of chr5 is a secondary hit that occurs shortly after the initial transforming event of MYC dysregulation (Fig. 5).

Reviewer 1, Comment 7)

I think that, if the authors wanted to link 5q loss and other mouse lesions to humans they should do it in an extensive and detailed way (e.g., listing genes). I personally don't feel that this link is needed in this paper, and I would consider removing that part (lines 234-240).

We believe there is value in attempting to translate the chr5 loss events to MM patients as it adds further interpretation of the chr5 loss finding, particularly because of the uneven distribution of mouse chr5 genes across human chromosomes (see **Supplementary Fig. 3d**). We have now listed the orthologous genes in **Supplementary Table 8**.

[Lines 228-230]: To gain insight into how loss of chr5 may translate to human myeloma patients, we performed mouse-to-human mapping of orthologous genes (Supplementary Fig. 3d, Supplementary Table 8).

Reviewer 1, Comment 8)

Is the GCN2 pathway activity correlated with the size of the M-protein?

We would not expect GCN2 pathway activity to correlate with M-protein size given that GCN2 pathway activity is highest in mice with early-MM (**Fig. 4c**). However, we have since conducted this analysis out of interest to the authors and reviewer, which found that the correlation between ISR-GCN2 activity and M-protein is not significant (see **Revision Fig. 4**, $R=0.415$, $P=0.0975$).

Reviewer 1, Comment 9)

How did the Author define "focal gain" (page 8, line 213)?

CNVs were considered focal if they were subchromosomal. As such, the authors have revised the manuscript text to be more technical in their reporting of results.

[Lines 206-207] CNVs not previously reported in this model included subchromosomal losses in chr3, chr9 and chr17 and subchromosomal gains in chr15.

Reviewer #2 (Expert in single-cell RNA-seq and cancer genomics): Remarks to the Author

In “Longitudinal single-cell analysis of a myeloma mouse model identifies early emergence of subclonal molecular programs associated with progression”, the authors investigated tumour progression of myeloma using longitudinal single-cell transcriptomics data from a genetic mouse model. The analyses showed extensive genetic and transcriptional heterogeneity of the malignant cells. The main findings are a shared subpopulation across samples with a ISR-GCN2 program related to protein translation and amino acid deprivation and an early subclonal loss of chr5 present in all tumours.

The study was well designed, however the analyses are a bit superficial in some parts. The authors show extensive transcriptomic heterogeneity, but what the specific subpopulations/states are and what drives this heterogeneity and how this heterogeneity drives tumour progression is not investigated in detail.

We thank the reviewer for acknowledging the quality of our study design and for their helpful constructive comments. Our responses and additional analyses below should clarify the transcriptional states of specific subpopulations and how the dynamics of these programs change throughout progression, thereby further enhancing the manuscript.

Reviewer 2, Comment 1)

*The authors show that cell state heterogeneity arises largely independently of CNV variation which has also been shown previously in Ledergor *et al.* (Nature Medicine, 2018) and are suggesting that there are additional non-genetic mechanisms driving tumour progression, for example epigenetic programs or the microenvironment. This is very interesting, but the authors did not go further to investigate the mechanisms that drive this heterogeneity.*

We agree that future studies should aim to delineate other sources that drive transcriptional heterogeneity, and believe that emerging single-cell multiome and spatial technologies will aid in these efforts. Unfortunately, these methods were not available at the time of sampling. However, we are preparing a manuscript focused on the non-malignant cell compartment of V κ *MYC mice during disease progression that should provide insight into how the immune microenvironment is involved in progression.

Reviewer 2, Comment 2)

Related to this transcriptomic heterogeneity, they identify 3-13 clusters across patients, however, they do not go into more detailed analyses to investigate the transcriptomic heterogeneity. What are those subpopulations? Are there shared clusters across samples?

In the original submission, we performed a detailed analysis investigating transcriptomic heterogeneity in clusters from **Fig. 3b**, the pathways driving this heterogeneity, and whether it is shared across samples (**Fig. 4a**). However, to make it clear to readers that the transcriptional clusters from **Fig. 3b** are the same clusters explored in **Fig. 4a**, we have now added the following text to the Results section of the manuscript:

[Lines 263-271] To further explore drivers of heterogeneity in the transcriptional clusters defined in Fig. 3b, we combined DE and enrichment analysis to define cluster-specific pathways within V κ *MYC tumours (Supplementary Table 9). Significantly enriched pathway terms for each transcriptional cluster were then compared pairwise to all other cluster-specific pathway terms using a Jaccard Similarity Index, which revealed two distinct transcriptional programs with representative clusters from all seven active-MM mice (Fig. 4a): Similarity Program A (11 clusters, mean Jaccard Index = 0.641) and Similarity Program B (9 clusters, mean Jaccard Index

= 0.660). The remaining 21 clusters displayed limited similarity (mean Jaccard Index=0.054), supporting that transcriptional divergence is a characteristic feature of disease progression.

In the original submission, we focused on defining pathways that had representative clusters from all active-MM samples (**Fig 4a**: Similarity Program A, Similarity Program B), but did not go into detail about the remaining clusters, as outlined by the Reviewer. This includes clusters previously defined as “Dissimilar”, which we have re-named “Divergent” in **Fig. 4a** to reflect that they are driven by programs not identified in all mice with active-MM. We then used Reactome enrichment analysis to explore the molecular programs associated with the groups of divergent clusters (**Revision Fig. 5**), the results of which are displayed in **Revision Fig. 2**. To provide readers with these additional insights, we have added **Revision Fig. 2** to the manuscript as **Supplementary Fig. 4**. We have also articulated these findings in the Results section:

[Lines 271-277] As depicted in Supplementary Fig. 4, the molecular processes associated with these divergent clusters included the immune system (innate immunity, cytokine signaling), cellular responses to stress (heat stress, metal ions), and signal transduction through mediators such as MAP kinase, nuclear receptors, and WNT. Thus, in addition to CNV-level events, transcriptional heterogeneity in the malignant compartment of V κ *MYC mice is also driven by other mechanisms, which likely includes external contributions from the tumour microenvironment.

Are there clusters enriched in early or late samples, are there specific cellular states promoting tumour progression?

Our data support that Similarity Program A (ISR-GCN2), which is expressed in a subpopulation of malignant cells from all V κ *MYC mice with active-MM, is enriched in malignant cells from early-MM mice (**Fig. 4c**). This observation was discussed in the Results section of the original submission:

[Lines 314-317] Not only was the ISR-GCN2 pathway upregulated in malignant cells from all disease groups compared to normal plasma cells, the highest level of ISR-GCN2 activation was observed in malignant cells from mice with the earliest stages of disease (early-MM) (Fig. 4c).

As outlined in our response to **Reviewer 1, Comment 2**, further analysis of the data supports that genes unregulated in malignant cells from int-MM mice (**Fig. 2e**) are highly expressed as marker genes in clusters that make up Divergent Program C in active-MM mice (new **Supplementary Fig. 5h**). Taken together, these findings support a model of disease progression where molecular variability that emerges in advanced disease (i.e. active-MM mice) may reflect residual programs from precursor disease that were selected against during progression. We created a new schematic that illustrates this model of progression supported by our data and included it as **Fig. 5** in the main manuscript.

Does the diversity increase or decrease over progression?

If we define diversity as intratumoural heterogeneity driven by subclonal CNVs, our data support that diversity remains stable throughout progression (**Supplementary Fig. 3b**). This is consistent with the “static model” of progression proposed by Bolli *et al.* as conferred in the Discussion section of our manuscript [**Lines 452-455**].

Our inferred CNV analysis supports the presence of distinct malignant cell populations at all stages of disease and further shows that these subclones persist throughout disease progression. This pattern of progression is consistent with the “static model” recently proposed by Bolli *et al.*⁶

In this model, subclonal architecture is retained as disease progresses from SMM to MM, such that progression reflects the time it takes to accumulate sufficient disease burden, rather than the acquisition of an additional oncogenic hit.

We did not directly establish diversity as defined by gene expression variability in the early-MM and int-MM disease stage groups because our estimation of intratumoural heterogeneity would be confounded by the small number of cells profiled from mice at these time points. We acknowledge this limitation in the Discussion section of our manuscript [**Lines 449-450**]:

We acknowledge that the small cell numbers profiled from mice with earlier disease confounds our estimation of intratumoural heterogeneity at these time points.

Since cell numbers were low at these earlier time points by virtue of disease burden being inherently lower, future studies can overcome this limitation by enriching for malignant cells prior to scRNA-seq profiling. We have added a statement to this effect in the Discussion [**Lines 450-452**]:

This limitation can and should be addressed in future studies by experimentally enriching for malignant cells prior to scRNA-seq profiling.

Reviewer 2, Comment 3)

If the progression is driven by non-genetic mechanisms, this could indicate plasticity. Are there continuous changes across tumour progression, are cells transitioning from one state to another during tumour progression? If yes, are there certain transcription factors driving this transition?

Our existing data set reveals that malignant cells demonstrate activation of the ISR-GCN2 pathway early in disease but then transition to a cell state related to immune processes before diversifying into several transcriptional subclones. This supports that states that exist in a particular disease stage persist into active disease. However, our experimental design doesn't enable this question to be answered directly since continuous changes and cell state transitions that occur throughout progression within an individual mouse would require serial sampling. We now acknowledge this in the Discussion:

[**Lines 409-415**] To better understand if and how these features contribute to differences in disease trajectories and rates of progression, future studies could consider exploring molecular changes that influence disease progression within an individual mouse via serial sampling. This approach could also enable the characterization of continuous changes such as cancer cell plasticity that may contribute to intratumoural heterogeneity and could further remove biological variability between mice that may have impacted our differential expression analysis across disease stages.

Reviewer 2, Comment 4)

The authors found only two shared programs, cell cycle and ISR-GCN2 program. Since the results do not show many DE genes between each stage of progression, another approach to look for shared molecular programs across samples would be using non-negative matrix factorisation and then looking how those programs/modules change along progression.

We thank the reviewer for this suggestion and have in turn performed further analyses of our dataset to identify shared molecular programs across samples using non-negative matrix factorization (NMF). Briefly, we applied NMF analysis to the malignant cells from all Vκ*MYC mice in our study and used

the top 50 features from each metagene factor for gene set enrichment analysis. As shown in **Revision Fig. 6a**, and consistent with our original analysis, two of the top metagene factors from NMF analysis were enriched for genes related to “Response of *EIF2AK4* (GCN2) to amino acid deficiency” (NMF8, FDR=0.0283) and “Cell cycle” (NMF9, FDR=1.021e-14) programs. NMF8 and NMF9 were also enriched with an overlapping set of terms identified in Similarity Program A (ISR-GCN2) and Similarity Program B (Proliferation), respectively (**Revision Fig. 6a, red boxes**).

The activity level of other top metagene factors detected by NMF were evaluated longitudinally and as shown in **Revision Fig. 6b**, NMF factors that change significantly along progression (NMF1, NMF2, NMF10, NMF20) similarly align to the peak-intermediate program identified as being elevated in int-MM mice by our longitudinal DE analysis (see **Fig. 2e**) and to Divergent Program C (see **Fig. 4a** and **Supplementary Fig. 4**). Thus, NMF analysis supports the findings derived from our original approach, but did not provide any additional insight into shared molecular programs across samples. For completeness, we have included this analysis as a new Supplemental Figure (**Supplementary Fig. 5d-e**) and referenced the analysis in the Results accordingly:

[Lines 306-310] Notably, a similar set of terms related to ISR-GCN2 was also detected using non-negative matrix factorization (Supplementary Fig. 5d), as were a separate set of terms related to proliferative processes akin to Similarity Program B described above (Supplementary Fig. 5e), further supporting the existence of these subclonal malignant programs in Vκ*MYC tumours.

Reviewer 2, Comment 5)

The validation using GCN2 inhibitor is convincing, but it would be good to see survival analyses using big cohorts like TCGA to support the prognostic relevance of the ISR-GCN2 pathway.

Unfortunately, multiple myeloma was not included in The Cancer Genome Atlas. However, we were able to conduct an analysis of bulk RNA-sequencing data using a large cohort of samples from the MMRF CoMMpass dataset (n=762 patients [insert ref]). Here, we did not identify a statistically significant difference in survival for patients with high vs. low ISR-GCN2 scores (**Revision Fig. 7**, p=0.25). We interpreted this to mean that either the ISR-GCN2 signature truly does not have prognostic significance in myeloma or that it is not clearly detectable in bulk RNA-seq data. Of these, the latter is supported by our analysis of single-cell myeloma data where the ISR-GCN2 signature is subclonal in a subset of myeloma patients (see specifically patients MM02 and MM09 from **Supplementary Fig. 6b**). Since large patient cohorts of scRNA-seq data are not yet available, we opted to exclude this analysis from the manuscript and instead focus on whether this pathway has therapeutic relevance.

Reviewer #3 (Expert in myeloma mouse models and therapy): Remarks to the Author

*In the manuscript by Croucher et al., the authors report a disease spectrum-spanning, single-cell transcriptional analysis of the V κ *MYC mouse model of myeloma, to study malignant cell evolution in the context of a uniform transforming path. Moreover, the authors show that the ISR-GCN2 pathway activity is progressively activated during progression in myeloma patients and is highly active in a subset of newly diagnosed MM samples. Indeed, chemical and genetic perturbation of GCN2 inhibits myeloma cell survival in vitro, which correlated with baseline ISR-GCN2 activity.*

The experiments are thoughtfully designed and the conclusions are supported by the results presented. However, I have some comments:

We thank the reviewer for acknowledging the quality of our study design and findings, and for their insightful suggestions to improve the experimental rigour of our manuscript. Our responses are below along with a description of results from the additional experiments requested.

Reviewer 3, Comment 1a)

In Figure 5, the authors show that the ISR-GCN2 pathway supports myeloma cell survival in vitro experiments, according to baseline ISR-GCN2 activity. In panel A, the authors show a progressive increase of ISR-GCN2 expression during progression in myeloma patients. The authors should evaluate if there is a direct functional interaction between MYC and GCN2. The authors should explore if MYC directly promotes GCN2 transcription.

The reviewer raises an important question about the interaction between MYC and the ISR-GCN2 pathway. Although our findings related to the ISR-GCN2 pathway were discovered in a model driven by MYC dysregulation, and while previous studies have described MYC-induced activation of the ISR-GCN2 pathway¹⁰, our data do not support a direct functional interaction (see also our response to Comment 1b). To make our interpretation of the data more clear, we have re-worded this section of the Results as follows:

[Lines 322-330] Given the role of MYC in promoting protein synthesis, these results put forth a model of disease in V κ *MYC mice whereby ISR-GCN2 activation occurs early in disease pathogenesis to tolerate cellular stress caused by MYC activation and the resulting excessive protein translation (Supplementary Fig. 5c). However, since MYC remains activated throughout V κ *MYC progression, the subsequent decrease in ISR-GCN2 activity in int-MM mice (Fig. 4c) and the subclonal nature of ISR-GCN2 activation in active-MM mice (Fig. 4a) suggests that GCN2 activation may be regulated by other mechanisms that drive amino acid deficiency. In myeloma, we speculate that this could be caused by excess production of immunoglobulin proteins, which is a hallmark physiological process in myeloma.

Since it is unknown whether MYC directly induces transcription of the gene encoding GCN2 (*EIF2AK4*) and in response to the reviewer's suggestion to explore if MYC directly promotes GCN2 transcription, we investigated whether MYC directly binds to regulatory elements in the *EIF2AK4* (GCN2) gene. As shown in **Revision Fig. 8**, chromatin immunoprecipitation (ChIP)-sequencing data from 3 separate cell lines confirms an enrichment of MYC transcription factor binding peaks within the 5' region of *EIF2AK4*. This therefore suggests that MYC can induce expression of *EIF2AK4* and that in early disease, this may be important to tolerate MYC activation/increased protein translation. However, we have not included this

finding in the revised manuscript since our data not support a correlation between MYC activity and ISR-GCN2 pathway activation (FIGURE). Moreover, activation of the ISR-GCN2 pathway is not driven by upregulation of *EIF2AK4*/GCN2 protein level, but rather by direct activation of GCN2 by uncharged tRNAs that result from amino acid deprivation.

Reviewer 3, Comment 1b)

The activity of GCN2-inhibitor does not seem related to MYC expression in tested MM cell lines. Indeed, the authors show no activity of GCN2-inhibitor in MYC overexpressing MMIS cells, while they demonstrate apoptosis induction in MYC-null U266 cells after GCN2-inhibitor. Furthermore, very modest activity is reported on Vκ12598 tumor cells. How the authors could explain these data?

We agree with the reviewer that the activity of GCN2iB does not appear related to MYC expression in the tested MM cell lines. To make this observation more empirical, we scored gene expression data from HMCLs using the ISR-GCN2 signature and a MYC transcriptional activity signature. As shown in **Revision Fig. 9**, MYC transcriptional activity scores were not correlated with ISR-GCN2 activity, nor were they correlated with GCN2iB response, suggesting that ISR-GCN2 pathway activation is independent of MYC in human myeloma. Rather, our data support that best predictor of GCN2iB response is ISR-GCN2 signature activation (**Fig. 6d**) regardless of MYC expression. These analyses have been added as Supplementary Figures (**Supplementary Fig. 7a-b**) in the revised manuscript and referenced in the text as follows:

[Lines 365-377] We therefore treated a panel of human myeloma cell lines (HMCLs, n=9) with the small molecule GCN2 inhibitor, GCN2iB,¹¹ to assess whether myeloma cells rely on the ISR-GCN2 pathway for survival. This analysis revealed that treatment with GCN2iB has anti-myeloma activity, but induced a range of responses in HMCLs (Fig. 6c), which we hypothesized may correlate with differences in basal ISR-GCN2 pathway activation. Indeed, by scoring publically-available HMCLs gene expression data using the ISR-GCN2 signature, we found a statistically significant inverse association between ISR-GCN2 score and viability after treatment with GCN2iB ($R=-0.605$, $P=0.028$, Fig. 6d), supporting that cell lines with higher ISR-GCN2 pathway activity are more sensitive to GCN2iB. Notably, MYC signature scores determined for each HMCL did not correlate with GCN2iB response ($R=-0.067$, $P=0.829$, Supplementary Fig. 7a) or ISR-GCN2 activity ($R=0.077$, $P=0.802$, Supplementary Fig. 7b), suggesting that ISR-GCN2 pathway activation may not be uniquely related to MYC dysregulation in human myeloma.

Finally, we have added a paragraph to the discussion to summarize our interpretation of the data as it relates to the relationship between MYC and ISR-GCN2 in myeloma.

[Lines 504-519] Given that MYC promotes global protein synthesis and drives malignant transformation in the Vκ*MYC model, it is tempting to speculate that there is a functional interaction between MYC and GCN2 in myeloma. Indeed, previous studies have described MYC-induced activation of the ISR-GCN2 pathway.¹⁰ However, several of our findings support that activation of the ISR-GCN2 pathway is not directly regulated by MYC in the Vκ*MYC model and human myeloma. For instance, constitutive activation of the MYC transgene in Vκ*MYC mice is the transforming event and thus, present in all cells. However, we observed low or subclonal levels of ISR-GCN2 activation in malignant cells from mice with int-MM and active-MM, respectively. Similarly, our *in vitro* studies revealed that activation of ISR-GCN2 in HMCLs was not correlated with MYC signature scores and that response to GCN2 inhibition was dependent on baseline ISR-GCN2 activation, not MYC activity. We suspect that the high levels

of ISR-GCN2 activity in V \$\kappa\$ *MYC mice with early disease reflects an adaptive mechanisms employed by myeloma cells to cope with cellular stress associated with transformation, but whether this is directly mediated by MYC has yet to be determine. Nonetheless, given the demonstrated importance of MYC dysregulation in myeloma biology and progression¹²⁻¹⁶, future studies should seek to better understand the interplay between MYC and GCN2 in this disease.

Regarding the reviewer's comment related to the modest activity of GCN2iB in the V κ 12598 tumour cells, these cells are highly sensitive to GCN2 inhibition, showing a 70.8% decrease in tumour cell survival after treatment with GCN2iB for 48 hours (**Fig. 6f**). To highlight this, we have delineated the quadrant of interest in **Fig. 6f** (i.e. CD138⁺B220⁻ myeloma cells) and updated the corresponding figure legend so that the results underpinning our interpretation are more clear.

Reviewer 3, Comment 1c)

The authors should check the effect of GCN2 knockdown in GCN2-i insensitive MM1S cell survival.

The reviewer proposes an important and valuable validation experiment and thus we generated GCN2 CRISPR knockouts using two GCN2-insensitive cell lines, MM1S and XG7. As shown in **Revision Fig. 10**, GCN2 CRISPR knockout did not have an effect on myeloma cell growth compared to control in either of these cell lines. We have included this cross-validation experiment in **Supplementary Fig. 7e-f** and in the Results section as follows:

[Lines 379-382] Consistent with these effects, genetic **knockout** of GCN2 in **GCN2iB-sensitive cell lines**, RPMI-8226 and OPM2, reduced cell viability (Fig. 6g, Supplementary Fig. 7e), but had no effect on viability of GCN2iB-insensitive cell lines, MM1S and XG7 (Supplementary Fig. 7e-f).

Reviewer 3, Comment 2)

The authors report the activation of processes related to chromosomal instability in cells with chr5 deletion as compared to wild-type chr5 (chr5WT) cells. Could the authors evaluate the differential occurrence of specific mutational signatures (PMID: 23945592) in chr5del vs chr5WT, to understand the potential role of error-prone DNA repair pathways, such as Alternative Non-Homologous End Joining Repair, in chr5del malignant plasma cells?

Although linking mutational signatures to our single-cell RNA-sequencing data would be extremely interesting, these methods were developed for and would require whole exome/whole genome sequencing data. Unfortunately, the samples described in our study were only profiled using scRNA-seq, and we did not have sufficient material for whole exome/whole genome sequencing.

References

1. Storti, P. *et al.* The transcriptomic profile of CD138(+) cells from patients with early progression from smoldering to active multiple myeloma remains substantially unchanged. *Haematologica* **104**, e465–e469 (2019).
2. Diamond, B. *et al.* Positive selection as the unifying force for clonal evolution in multiple myeloma. *Leukemia* 1–5 (2021). doi:10.1038/s41375-021-01130-7
3. Fan, J. *et al.* Linking transcriptional and genetic tumor heterogeneity through allele analysis of single-cell RNA-seq data. *Genome Research* **28**, 1217–1227 (2018).
4. Oben, B. *et al.* Whole-genome sequencing reveals progressive versus stable myeloma precursor conditions as two distinct entities. *Nature Communications* 1–11 (2021). doi:10.1038/s41467-021-22140-0
5. Rustad, E. H. *et al.* Timing the initiation of multiple myeloma. *Nature Communications* 1–14 (2020). doi:10.1038/s41467-020-15740-9
6. Bolli, N. *et al.* Genomic patterns of progression in smoldering multiple myeloma. *Nature Communications* **9**, 1870–10 (2018).
7. Broad Institute Group (data production and analysis). inferCNV of the Trinity CTAT Project. (2019).
8. Chesi, M. *et al.* Monosomic Loss of MIR15A/MIR16-1 Is a Driver of Multiple Myeloma Proliferation and Disease Progression. *Blood Cancer Discov* 1–14 (2020). doi:10.1158/0008-5472.BCD-19-0068
9. The Murine V_k_MYC Myeloma Shares Defining Genetic Lesions with Human Multiple Myeloma. 1–3 (2017).
10. Tameire, F. *et al.* ATF4 couples MYC-dependent translational activity to bioenergetic demands during tumour progression. *Nat Cell Biol* **21**, 889–899 (2019).
11. Nakamura, A. *et al.* Inhibition of GCN2 sensitizes ASNS-low cancer cells to asparaginase by disrupting the amino acid response. *Proceedings of the National Academy of Sciences* **115**, E7776–E7785 (2018).
12. Shou, Y. *et al.* Diverse karyotypic abnormalities of the c-myc locus associated with c-myc dysregulation and tumor progression in multiple myeloma. *Proceedings of the National Academy of Sciences* **97**, 228–233 (2000).
13. Affer, M. *et al.* Promiscuous MYC locus rearrangements hijack enhancers but mostly super-enhancers to dysregulate MYC expression in multiple myeloma. *Leukemia* **28**, 1725–1735 (2014).
14. Avet-Loiseau, H. *et al.* Rearrangements of the c-myc oncogene are present in 15% of primary human multiple myeloma tumors. *Blood* **98**, 3082–3086 (2001).
15. Misund, K. *et al.* MYC dysregulation in the progression of multiple myeloma. *Leukemia* 1–5 (2019). doi:10.1038/s41375-019-0543-4
16. Keane, N. *et al.* MYC Translocations Identified By Sequencing Panel in Smoldering Multiple Myeloma Strongly Predict for Rapid Progression to Multiple Myeloma. *Blood* **130**, 393 (2017).

Revision Figures

Longitudinal single-cell analysis of a myeloma mouse model identifies early emergence of subclonal molecular programs associated with progression

Revision Figures 1-10

Figure R1 | Detection of dominant transcriptional program from int-MM mice in subclones of active-MM mice. Gene set scoring across transcriptional clusters for “peak int-MM genes” signature developed from analysis in Fig. 2e. Scores were calculated using Seurat’s AddModuleScore. Results are presented for AMM1, AMM3 and AMM4 as boxplots within violin plots to depict the distribution of each measurement within clusters (central rectangle spans the interquartile range, central line represents the median, and “whiskers” above and below the box show the value 1.5x the IQR). Clusters with the highest scores (AMM1_C9, AMM1_C7, AMM3_C4, AMM4_C8) correspond to clusters in Divergent Program C from Fig 4a.

Figure R2 | Subclonal transcriptional programs in $V\kappa^*$ MYC mice with active-MM mice. Map of reactome terms with significant enrichment in malignant cell clusters from Divergent programs defined in Fig 4a (associated clusters are listed next to Program name). The full hierarchy of each reactome pathway is shown for context but only significantly enriched shared pathways are highlighted in orange.

Figure R3 | Effect of cell cycle stages on mapping CNV and gene expression data. (a) UMAP visualization of malignant cells from representative active-MM mouse data sets coloured by transcriptional cluster. **(b)** UMAP visualization of malignant cells from representative active-MM mouse data sets coloured by inferCNV subpopulation. **(c)** UMAP visualization of malignant cells from representative active-MM mouse data sets coloured by relative expression of *Mki67* to highlight proliferative cluster. **(d)** Barplot showing the distribution of cell cycle phase across transcriptional clusters (x-axis). **(e)** Barplot showing the distribution of CNV subpopulations (fill) across transcriptional clusters (x-axis) separated into two plots for proliferative clusters vs. non-proliferative clusters. Results for a-e are organized for active-MM mouse AMM1, AMM2, and AMM7 in columns, with subject names and number of cells/transcriptional clusters listed above.

Figure R4 | Pearson correlation between size of M-protein (x-axis) and GCN2 pathway activity (y-axis). The linear regression line is plotted in black with confidence interval shaded grey. Each dot represents one mouse, coloured by disease stage group.

Figure R5 | Grouping analysis of Divergent Programs in active-MM mice. Distinct transcriptional clusters were grouped into Divergent Programs C-L using k-means clustering as implemented in ComplexHeatmap.

Figure R6 | NMF analysis recapitulates findings from longitudinal DE analysis. (a) Top 20 enriched terms from Reactome analysis ($FDR \leq 0.05$) of NMF metagene factors NMF8 and NMF9 (top 50 feature loadings), which align with shared pathways identified in Fig 4a of main manuscript (red boxes). (b) Top enriched terms (maximum 20) from Reactome analysis ($FDR \leq 0.05$) of NMF metagene factors that changed significantly throughout progression. Disease stage-specific scoring of corresponding NMF metagene factors is shown to the right of enrichment plots. In these line plots, dots represent mean NMF score of disease stage samples, which was derived by scoring (Seurat::AddModuleScore) malignant cells for the top 50 feature loadings of a given NMF metagene factor. Error bars depicting standard error of the mean. Statistical comparisons were performed using a two-sided t-test with subsequent correction for multiple testing (Bonferroni, $*P \leq 0.05$).

Figure R7 | Survival analysis for ISR-GCN2 signature. (a) KM plot for patients with high ISR-GCN2 gene set score compared to patients with low ISR-GCN2 gene set score (log-rank test, two-sided).

Figure R8 | MYC transcription factor binding sites in *EIF2AK4* (GNC2) gene. IGV snap shot of ChIP-seq peaks from (a) GM12878, a human EBV-transformed lymphoblastoid cell line (ENCODE), and (b) MM1S, a human MYC overexpressing MM cell line (GSE36354, PMID:23021215), and (c) K562, a human chronic myelogenous leukemia cell line (ENCODE).

Figure R9 | Relationship between ISR-GCN2 and MYC in HMCLs. (a) Pearson correlation between MYC transcriptional activity score (x-axis) and viability relative to DMSO (1 μ M GCN2iB, y-axis). (b) Pearson correlation between MYC transcriptional activity score (x-axis) and GCN2 pathway activity (y-axis). In both (a) and (b), the linear regression line is plotted in black with confidence interval shaded grey and each dot represents one HMCL.

Figure R10 | GCN2 KO in GCN2iB-insensitive cell lines. (a) Western blot analysis of GCN2 CRISPR knockout in MM1S and XG7 myeloma cell lines. **(b)** Bar plots depicting cell survival, as determined by Trypan blue assay, in GCN2 knockout MM1S and XG7 myeloma cell lines.

REVIEWER COMMENTS

Reviewer #1 (Remarks to the Author):

I previously had a favorable opinion of this paper, except for a few minor comments that have been fully addressed in the new version.

Reviewer #2 (Remarks to the Author):

The reviewers have answered most of my comments and the manuscript has been considerably improved. However, I am still not completely convinced of the transcriptional heterogeneity, and whether this is not driven by potentially cell cycle stages or uncertainty in the clustering as suggested by the comment 5 from Reviewer 1. Looking from the Revision figure 3, it seems to me that many clusters are potentially driven by CNVs, if you don't consider the proliferating clusters. It is also surprising that the same genetic model leads to so many divergent transcriptional states across different mice and only two shared programs. From the additional analyses of the authors of the divergent clusters, it was not very clear what those clusters/cancer states are and in addition, this showed that one of those divergent programs is cellular responses to stress which might indicate for example upregulation of genes associated with dissociation effects (Brink et al., Nature Methods, 2017). To show that this doesn't affect the transcriptional heterogeneity not driven by CNVs, I would suggest to reanalyse the data by removing all cell cycle genes (for example using the G2M and S phase lists of genes in Seurat), the genes associated with dissociation effects (Brink et al., Nature Methods, 2017) and potentially also all mitochondrial and ribosomal genes (as in many different single-cell studies it has been shown that often there are clusters only differentiated by upregulation of either ribosomal or mitochondrial genes) and to see whether there is still such transcriptional heterogeneity divergent across different mice and not driven by CNVs. In addition, it would be great to provide a list of top20 or top10 markers (DE genes) for all clusters which would be helpful for readers to better understand the different clusters.

Reviewer #3 (Remarks to the Author):

The authors addressed all my comments

Response to Reviewers – Croucher et al.

We thank the reviewers for their latest comments on our manuscript. We have responded in-kind to address the revisions proposed by Reviewer #2. Reviewer comments are in **red, italicized text**, followed by our response in black text and any resulting manuscript changes (verbatim) in **blue, underlined text** (blue, non-underlined text may be provided to show context of the change within the revised manuscript). Additional figures supporting our response can be found at the end of this document.

Reviewer #2 (Expert in single-cell RNA-seq and cancer genomics): Remarks to the Author

Reviewer 2, Comment 1a)

The reviewers have answered most of my comments and the manuscript has been considerably improved. However, I am still not completely convinced of the transcriptional heterogeneity, and whether this is not driven by potentially cell cycle stages or uncertainty in the clustering as suggested by the comment 5 from Reviewer 1. Looking from the Revision figure 3, it seems to me that many clusters are potentially driven by CNVs, if you don't consider the proliferating clusters. It is also surprising that the same genetic model leads to so many divergent transcriptional states across different mice and only two shared programs.

In our interpretation of the data, it is not entirely surprising to have observed divergent transcriptional states in different V κ *MYC mice as, while each cancer is derived from a common oncogenic initiating event (MYC dysregulation), there are multiple additional paths to malignancy. Indeed, inter-tumoural heterogeneity has been reported in this model previously by Chesi *et al.*¹ using array-based comparative genomic hybridization (see Figure to right) applied to 26 tumours from independent V κ *MYC mice. These results provide an example of how the same genetic model can lead to divergent genomes. Although our data was not able to identify an underlying genetic driver of the two shared transcriptional programs, ISR-GCN2 and proliferation, we suspect that they provide a survival advantage and therefore would be recurrently selected for throughout progression.

*From the additional analyses of the authors of the divergent clusters, it was not very clear what those clusters/cancer states are and in addition, this showed that one of those divergent programs is cellular responses to stress which might indicate for example upregulation of genes associated with dissociation effects (Brink *et al.*, *Nature Methods*, 2017). To show that this doesn't affect the transcriptional heterogeneity not driven by CNVs, I would suggest to reanalyse the data by removing all cell cycle genes (for example using the G2M and S phase lists of genes in Seurat), the genes associated with dissociation effects (Brink *et al.*, *Nature Methods*, 2017) and potentially also all mitochondrial and ribosomal genes (as in many different single-cell studies it has been shown that often there are clusters only differentiated*

by upregulation of either ribosomal or mitochondrial genes) and to see whether there is still such transcriptional heterogeneity divergent across different mice and not driven by CNVs.

We thank the reviewer for their suggestion and for pointing out the interesting study by van den Brink *et al.*² Of note, their dissociation method is significantly different from what we used in our study. To summarize the van den Brink supplemental methods, tissues were dissected mechanically by razor blade, digested enzymatically with collagenase, passed through a pipette tip and then a 20 gauge needle. These cells were then passed through a cell strainer, stained with Hoechst and subject to FACS. Our protocol utilizes substantially less cellular manipulation, involving flushing of suspension cells from hind leg bone marrow, gentle dissociation by passing through a 25 gauge needle, and then immediately loading on the 10X Genomics Chromium for cell encapsulation. No FACS was used in our protocol. Further, myeloma is a disease where cells are under significant ER and mitochondrial stress from excessive immunoglobulin production and ROS.

However, we do agree that it is important to quantify any dissociation-associated signal in our data and pursued the analysis you recommended. Specifically, we have re-analyzed our data to assess whether transcriptional heterogeneity remains after removal of dissociation-associated (Supplementary Table 5 from van den Brink *et al.*²), cell cycle, mitochondrial, and ribosomal genes. Although the removal of these genes did change the final clustering solutions by producing a different number of clusters (see **Revision Fig. 1a** vs. **Revision Fig. 1b**), we still observed transcriptional heterogeneity in the malignant cell compartment independent of CNV status (**Revision Fig. 1c-d**). Using AMM3 as an example, clusters 1 and 2 remain transcriptionally distinct even after removal of the genes in question, despite displaying the same CNV profile (green, subpop_3) (**Revision Fig. 1d**). Most importantly, the shared ISR-GCN2-driven subpopulation observed in the original analysis was still present after removal of these potentially confounding genes (see starred clusters in **Revision Fig. 1a-b**). These patterns were similar across all mice, except AMM5 and AMM7, which might be explained by these samples having the lowest cell counts (302 and 310 cells, respectively). Thus, our re-analysis supports the robustness of our original analysis that found transcriptional heterogeneity in the malignant compartment of V κ *MYC mice. Since this point may be of interest to readers, we have added the re-analysis from **Revision Fig. 1** as **Supplementary Fig. 4** in our manuscript and revised the main text as follows:

[Lines 246-259] We then evaluated whether this transcriptional heterogeneity was driven by the subclonal CNVs inferred above by exploring the distribution of CNV-driven subpopulations within each transcriptional cluster. In doing so, we observed instances of majority CNV-driven transcriptional clusters (Fig. 3c-d). For example, in AMM1, transcriptional cluster 4 was largely comprised of cells from CNV subpopulation 3, defined by del(5) and del(12). Similarly, in AMM4, transcriptional cluster 0 was largely comprised of cells from CNV subpopulation 1, defined by subchromosomal gain of chromosome 9. This supports that subclonal CNVs can have a significant effect on the formation of distinct transcriptional clusters. However, the majority of CNV-driven subpopulations were distributed across several transcriptional clusters (Fig. 3d) and we did not find a significant correlation between the number of CNV-driven subpopulations and the number of transcriptional clusters ($R=-0.597$, $P=0.1567$, Supplementary Fig. 3e). Moreover, this transcriptional heterogeneity was retained when potentially confounding genes associated with dissociation³⁸ and mitochondrial/ribosomal/cell cycle genes were removed (Supplementary Fig. 4). Thus, our data support that transcriptional variability must be driven by additional sources beyond subclonal CNV events.

Reviewer 2, Comment 1b)

In addition, it would be great to provide a list of top20 or top10 markers (DE genes) for all clusters which would be helpful for readers to better understand the different clusters.

We agree with the Reviewer and have added this information to the manuscript as a new Supplementary Table (**Supplementary Table 10**), with reference in the main text as follows:

[Lines 265-267] To further explore drivers of heterogeneity in the transcriptional clusters defined in Fig. 3b, we combined DE and enrichment analysis to define cluster-specific pathways within V κ *MYC tumours (Supplementary Tables 9-10).

Revision Figures Round #2

Longitudinal single-cell analysis of a myeloma mouse model identifies early emergence of subclonal molecular programs associated with progression

Revision Figure 1

Figure R1 | Impact of removing dissociation, mitochondrial, ribosomal and cell cycle genes on transcriptional heterogeneity. (a) UMAP visualization of malignant cells from each active-MM mouse coloured by original transcriptional cluster. **(b)** UMAP visualization of malignant cells from each active-MM mouse coloured by revised transcriptional clusters upon removal of dissociation, mitochondrial, ribosomal, and cell cycle genes. After removal of these genes, data were re-analyzed using a multi-resolution clustering approach as described in Methods. The resulting clusters were then mapped back to UMAP plots generated from (a) for comparison. **(c)** UMAP visualization of malignant cells from each active-MM mouse coloured by CNV subpopulation. **(d)** Bar plot showing the distribution of CNV subpopulations (fill) across revised transcriptional clusters (x-axis). Results are organized for each active-MM mouse in columns, with subject names and number of cells/transcriptional clusters listed above.

References

1. Chesi, M. *et al.* Monosomic Loss of MIR15A/MIR16-1 Is a Driver of Multiple Myeloma Proliferation and Disease Progression. *Blood Cancer Discov* 1–14 (2020). doi:10.1158/0008-5472.BCD-19-0068
2. van den Brink, S. C. Single-cell sequencing reveals dissociation-induced gene expression in tissue subpopulations. *Nat Meth* **14**, 935–936 (2017).

REVIEWERS' COMMENTS

Reviewer #2 (Remarks to the Author):

The authors addressed all my comments